# Ordered arrangement of dendrites within a *C. elegans* sensory nerve bundle

Zhiqi Candice Yip[1,2], Maxwell G Heiman[1,2]*

[1]Division of Genetics and Genomics, Boston Children's Hospital, Boston, United States; [2]Department of Genetics, Harvard Medical School, Boston, United States

**Abstract** Biological systems are organized into well-ordered structures and can evolve new patterns when perturbed. To identify principles underlying biological order, we turned to *C. elegans* for its simple anatomy and powerful genetics. We developed a method to quantify the arrangement of three dendrites in the main sensory nerve bundle, and found that they exhibit a stereotyped arrangement throughout larval growth. Dendrite order does not require prominent features including sensory cilia and glial junctions. In contrast, loss of the cell adhesion molecule (CAM) CDH-4/Fat-like cadherin causes dendrites to be ordered randomly, despite remaining bundled. Loss of the CAMs PTP-3/LAR or SAX-7/L1CAM causes dendrites to adopt an altered order, which becomes increasingly random as animals grow. Misexpression of SAX-7 leads to subtle but reproducible changes in dendrite order. Our results suggest that combinations of CAMs allow dendrites to self-organize into a stereotyped arrangement and can produce altered patterns when perturbed.
DOI: https://doi.org/10.7554/eLife.35825.001

## Introduction

Biological systems reflect a remarkable hierarchy of structural organization: proteins assemble into nanometer-scale machines, that in turn build cells, that are then organized and patterned to form structures as complex as the human brain. How do these pieces come together in predictable ways? From first principles, one can imagine a top-down deterministic approach in which the shape, position, and connections of each biological part are specified, similar to an electrical wiring diagram or an architectural blueprint. While this approach is reasonable for simple systems, it breaks down quickly in the face of the complexity we encounter in biology. For example, the human brain contains over 100 billion neurons and glial cells (*Herculano-Houzel, 2009*), all of which are precisely connected to form a functioning organ. It is hard to imagine a top-down program that can reliably produce such well-organized brains across a population of individuals.

A powerful alternative strategy for organization is a bottom-up, rules-based approach which has been widely considered in studies of morphogenesis and pattern formation. This type of approach has been called 'emergent' organization, because well-ordered patterns emerge from a series of local interactions rather than being specified by a blueprint. For example, Alan Turing's reaction-diffusion mechanism, which consists of an activator molecule that can make more of itself, an inhibitor molecule that inhibits production of the activator, and a mechanism for diffusing these two molecules (*Turing, 1952*), has been suggested to create the diverse array of biological patterns found in fish stripes (*Kondo and Asal, 1995*), in seashells (*Meinhardt, 1995*), and in fur and skin coloration such as those of giraffes and lizards (*Manukyan et al., 2017*; *Walter et al., 1998*). A second example of a rule that can generate biological patterns is differential adhesion, in which biological parts sort themselves based on adhesivity to create elaborate structures and patterns. This concept was first demonstrated in 1955 by Townes and Holtfreter, who dissociated different germ layers of embryonic amphibian cells, mixed them together, and found that these cells re-aggregated and

*For correspondence:
heiman@genetics.med.harvard.edu

**Competing interests:** The authors declare that no competing interests exist.

sorted out into layers according to their cell type (*Townes and Holtfreter, 1955*). Differential adhesion has also been shown to organize nerve bundles in the *Drosophila* visual system (*Schwabe et al., 2014*). From an evolution standpoint, these emergent, bottom-up strategies may be more advantageous than following a global blueprint, as changes to the starting conditions do not necessarily lead to a disorganized jumble, but instead can give rise to novel well-ordered patterns. For example, we recently showed how altering cell number in the *C. elegans* nervous system gives rise to an emergent, well-ordered pattern of dendrite arbors (*Yip and Heiman, 2016*).

To identify rules that help to organize cells into other kinds of ordered structures, we decided to study a simple, stereotyped structure in the *C. elegans* nervous system called the amphid sense organ. The amphid contains 12 sensory neurons and two glial cells. Each neuron extends a single unbranched dendrite that terminates at the nose tip in a sensory cilium that senses environmental stimuli (*Ward et al., 1975*). Together, these dendrites and their associated glial processes fasciculate to form one of four bilaterally symmetric nerve bundles that constitute the sensory structures of the head. These structures were the first portion of the *C. elegans* nervous system to be reconstructed by electron microscopy (*Ward et al., 1975*). Based on analysis of four animals, Ward and colleagues noted that amphid dendrites appeared to be ordered consistently within the bundle relative to one another, commenting that 'Although individual worms are not precise replicas of each other down to the finest details, they are remarkably exact copies' (*Ward et al., 1975*). These observations are also consistent with recent EM (*Doroquez et al., 2014*). Together, the studies show that amphid bundles are remarkably well ordered, yet the problem of how dendrite order arises has not been pursued, largely because of the painstaking methods required to examine it.

In addition to the evidence from these EM studies, we chose to study dendrite order in the amphid for four reasons. First, the question of how axons and dendrites are organized within bundles is relatively unexplored, yet a neuron's choice of neighbors may affect its developmental outgrowth, mature activity, or susceptibility to damage or age-related disease. Second, the amphid bundle is a relatively simple and well-isolated system compared to other bundles of processes in the nervous system, containing only 12 unique and identifiable dendrites with no gap junctions or synapses between them (*Doroquez et al., 2014*; *Ward et al., 1975*). Third, the system allows us to easily distinguish defects in fasciculation versus guidance. Amphid dendrites do not grow outwards to the nose from a stationary cell body, but instead form by anchoring to the embryonic nose tip while the cell bodies migrate away together, a phenomenon known as retrograde extension (*Heiman and Shaham, 2009*). Thus, amphid dendrite fasciculation can be studied independently of outgrowth and guidance. Finally, promoters with single-cell-specificity are readily available for all amphid neurons, allowing us to easily visualize and genetically manipulate any single amphid neuron in live animals.

## Results

### Development of a semi-automated method to quantify dendrite order

To study the cellular and molecular mechanisms underlying dendrite order in the amphid, we first sought to develop methods to measure, quantify, and compare dendrite order in populations of animals. Ideally, we would label each of the 12 amphid dendrites with a unique fluorescent marker, examine cross sections of the dendrite bundle, and determine how dendrites are ordered across individuals. However, we lacked the technical ability to generate 12 differently colored cell-specific markers and, even if we could uniquely mark each neuron, quantifying the degree of stereotypy of 12 dendrites is a mathematically complex problem. Thus, we decided to simplify the problem by labeling only three dendrites and examining their relative order as a proxy for overall bundle order. We reasoned that, if the overall bundle were well-ordered, then any three dendrites would be similarly well-ordered. Our method can be broken down into three parts: imaging, quantification, and statistical analysis (*Figure 1*).

First, we generated a strain to label three amphid neurons in different colors (AWA:YFP, AFD: CFP, ASE:mCherry) by combining two separately integrated transgenes (one with AWA:YFP, the second with AFD:CFP and ASE:mCherry) (*Figure 1A*). We used this YFP + CFP/mCherry approach because we consistently observed recombination artifacts when CFP and YFP were introduced on the same transgene. Using traditional widefield deconvolution microscopy, we found that dendrites in newly hatched larvae were too close together to be reliably resolved (*Figure 2—figure*

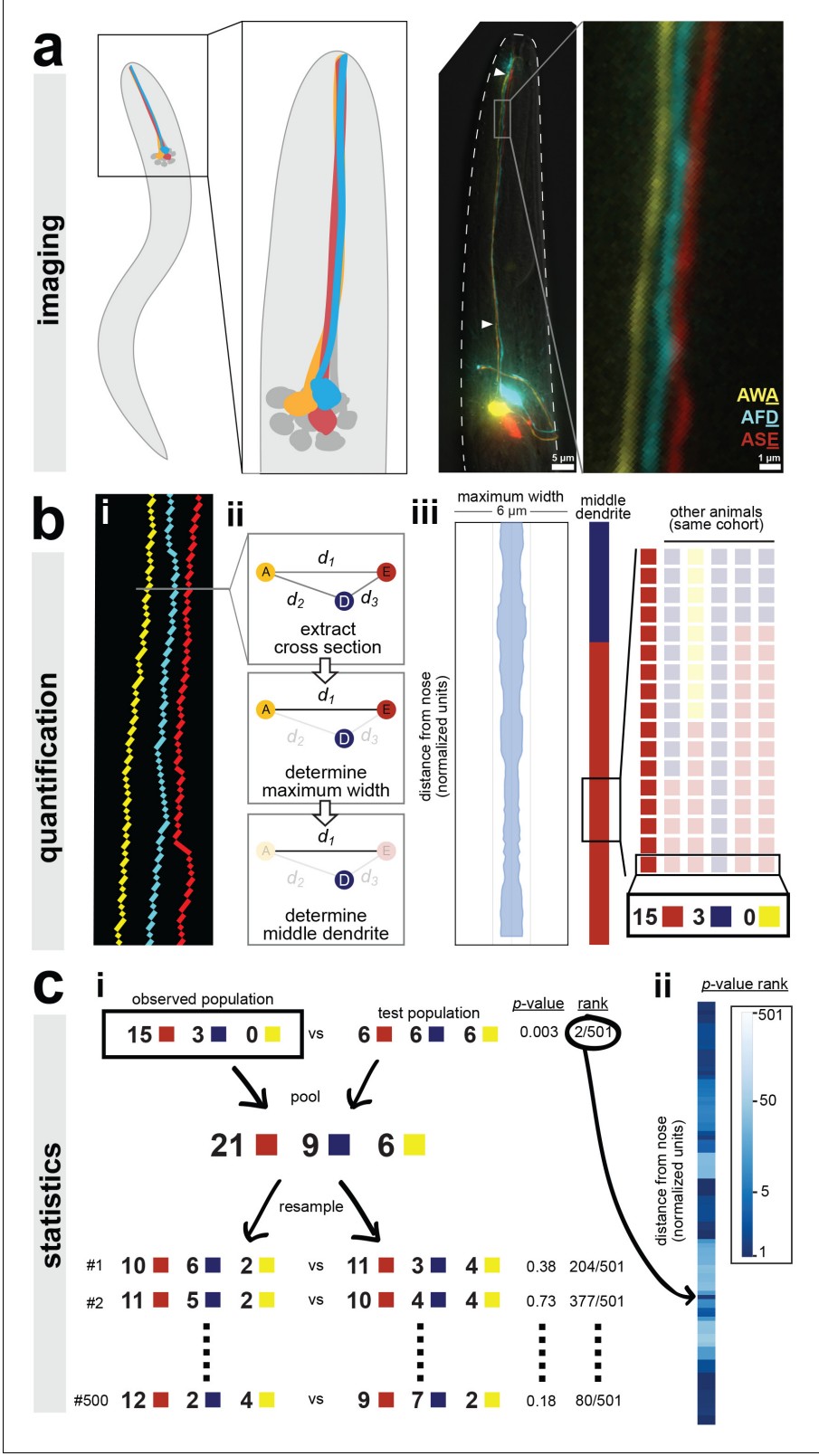

**Figure 1.** A semi-automated method for quantifying dendrite order. Our analysis pipeline can be divided into three parts: (A) imaging, (B) quantification, and (C) statistics. (A) Schematic showing position of amphid neurons in head and maximum-intensity projection of a fourth larval stage (L4) animal expressing YFP, CFP, and mCherry in AWA, AFD, and ASE, respectively. Nose tip is at top. Arrowheads indicate approximate extent of bundle used

*Figure 1 continued on next page*

*Figure 1 continued*

for analysis in all experiments. (**B**) (i) To quantify dendrite order, a brightest-path algorithm first extracts the 3D coordinates of each dendrite. (ii) In cross-section, the dendrites define a triangle (A, AWA; D, AFD; E, ASE). The pairwise distances between dendrites are determined ($d_1$, $d_2$, $d_3$). The longest distance is taken as a proxy for bundle width ($d_1$ in the schematic). The dendrite opposite the longest side is defined as the dendrite 'in the middle' of the others (AFD in the schematic). (iii) For visualization, bundle width is plotted at each position along the bundle. The dendrite in the middle is plotted as a color bar (yellow, AWA; blue, AFD; red, ASE). Color bars from age- and genotype-matched individuals are arranged side-by-side to visualize patterns of dendrite order in a population. For simplicity, only n = 6 bundles are illustrated, yielding a distribution at the boxed position of 5 red, 1 blue, 0 yellow; for a typical sample size of n = 18, these would correspond to 15 red, 3 blue, 0 yellow. (**C**) (i) For statistical testing, the observed distribution (e.g. 15, 3, 0) is compared to a test distribution using Fisher's $3 \times 2$ exact test as a test statistic to obtain a nominal 'true' *p*-value. Permutation testing is carried out by merging observed and test distributions and resampling 500 times, to obtain representative p-values for distributions with the same composition that differ only by sampling error. The 'true' *p*-value is ranked relative to these resampled *p*-values, with lower rankings (1/501) indicating the true distributions differ more than would be expected by sampling error. Rankings below 25/501 or 5/501 represent a corrected *p*-value of p<0.05 or p<0.01 respectively. (ii) This comparison is carried out at every position along the bundle and the *p*-value rankings are represented as a log-scale color bar. Pairwise distances, triangles, and *p*-value rankings at every position along each bundle in this study can be explored at http://heimanlab.com/ibb.
DOI: https://doi.org/10.7554/eLife.35825.002

*supplement 1A*) but dendrites in older larvae and adults were readily discerned despite each dendrite being only about 0.5 µm in diameter, close to our effective resolution limit (*Figure 1A* and *Figure 2—figure supplement 1A*) (*Doroquez et al., 2014*; *Ward et al., 1975*).

Next, we sought to quantify the order of dendrites within individual animals (*Figure 1B*). To do this, we used a custom MATLAB script that extracts the x-, y-, and z-coordinates of the three dendrites, calculates a centroid line that runs between the three dendrites, and, for each point along the centroid, finds the point on each of the three dendrites that is closest to the centroid (*Figure 1Bi*). These three dendrite points define a single cross section (*Figure 1Bii*). Thus, independent of any curvature or rotation in the head of a given animal, this approach identifies a series of contiguous cross-sections along the length of the dendrite bundle that intersect the three dendrites exactly once. Each cross section contains a triangle consisting of one ASE, AFD, and AWA dendrite point. We calculated the pairwise distances between these points (*Figure 1Bii*).

We used these three pairwise distances to quantify dendrite order in two ways. First, we used the longest pairwise distance as a proxy for bundle width, allowing us to measure how tightly or loosely the dendrites are bundled. To do this, we simply plotted the value of the longest pairwise distance at every point along the bundle (*Figure 1Biii*). For example, for the animal shown in *Figure 1*, the longest pairwise distance at every point along the length of the bundle is less than 2 µm, suggesting that the dendrites in this animal are tightly bundled together (*Figure 1Biii*). Second, we created a categorical variable that describes relative dendrite order by identifying the dendrite 'in the middle' of the others at every point along the bundle. A dendrite is considered to be in the middle if it is opposite the longest side of the triangle, that is, the longest pairwise distance. For example, if the longest pairwise distance is $d_1$, which is the distance between AWA (*Figure 1Bii*, yellow 'A') and ASE (*Figure 1Bii*, red 'E'), then the dendrite in the middle at that point along the bundle is AFD (*Figure 1Bii*, blue 'D'). Importantly, this method describes the arrangement of dendrites relative to each other within the bundle, and is not affected by rotation or twisting of the bundle or head. Next, to visualize dendrite order along the length of the bundle in a single animal, we simply plotted the color of the middle dendrite at each position along the bundle in a column (*Figure 1Biii*). For example, for the animal in *Figure 1*, the dendrite in the middle is AFD (blue) near the nose tip and then switches to ASE (red) closer to the cell bodies. To visualize dendrite order within a population, we generated a 'population plot' by arranging these individual columns side-by-side (*Figure 1Biii*).

Finally, we employed a statistical approach based on permutation tests to quantitatively compare the observed dendrite order in a population to a simulated random order (*Figure 1Ci*, see 'Statistics' in Methods for additional details including our rationale for choosing this approach). We also used this approach to compare dendrite order between wild-type and mutant populations. In this approach, our null hypothesis is that the two populations are drawn from the same distribution and

any differences between them merely reflect sampling error. First, the observed and test populations are compared using a test statistic (we used Fisher's exact test, see Methods) to yield a nominal *p*-value. These populations are then computationally merged and re-sampled to create mock populations, and the test statistic is recalculated. Repeating this permutation process (500 iterations) gives a representative set of *p*-values for populations that have the same composition as our true samples but, by definition, differ only by sampling error. This approach yields 501 *p*-values (1 true nominal p-value+500 *p*-values from resampling) for each point along the length of the dendrite bundle. The percentile rank of the true nominal *p*-value at each point is plotted using a log-scale color bar (*Figure 1Cii*). Darker colors indicate a low rank, meaning that the observed dendrite order is significantly different from the test distribution. The rank value is equivalent to a corrected *p*-value, for example rankings less than 25/501 are equivalent to p<0.05 and rankings less than 5/501 are equivalent to p<0.01.

The pairwise distances, dendrite triangles, and *p*-value ranks for all positions along each bundle in this study (n = 475 bundles) can be explored with a graphical interface using an 'interactive bundle browser' we created (http://heimanlab.com/ibb).

To summarize, we have developed a robust and semi-automated pipeline to detect, quantify, and compare dendrite order, which now allows us to determine dendrite order in a wild-type population and to ask whether it is altered by various manipulations.

## Amphid dendrites are fasciculated and ordered

To determine wild-type dendrite order, we imaged animals expressing CFP, YFP, and mCherry in AFD, AWA, and ASE respectively. Animals were synchronized at the first larval stage (L1) and then collected and imaged at three time points (24 hr, early larval stage (L2/3); 48 hr, late larval stage (L4); 72 hr, one-day adult). We found that dendrites are tightly fasciculated throughout larval growth (*Figure 2A*) (average longest pairwise distance ± standard deviation: 24 hr, 0.88 ± 0.09 μm; 48 hr, 1.11 ± 0.22 μm; 72 hr, 1.51 ± 0.25 μm). The approximate doubling in bundle width may reflect an increased diameter of each dendrite and thus the entire bundle, and roughly corresponds to the overall doubling in length of the head during these stages.

Next, we looked at dendrite order across the population (*Figure 2B,C*). We visualized dendrite order using population plots as described above (*Figure 2C*) as well as summary plots that represent the fraction of animals with the ASE, AWA, or AFD dendrite in the middle at every point along the bundle (*Figure 2B*). The summary plot provides a compact way to visualize how well-ordered dendrites are in a population. If all of the lines converge at 0.33 then it indicates that dendrite order is random for that population, whereas if any of the lines approach 1 or 0 then it suggests that dendrites are arranged in a consistent, non-random order.

We found that amphid dendrites are well-ordered, especially in younger animals (*Figure 2B,C*; left plots). To quantify the degree of order, we compared our observed samples to a simulated random sample using a permutation test (*Figure 2B*, blue bars) as well as a chi-squared test (*Figure 2—figure supplement 1B*, see Methods). Interestingly, although dendrite order is stereotyped, it is not uniform along the length of the bundle. Instead, it exhibits a switch point, which itself is stereotyped (*Figure 2B,C*). Close to the nose tip, AFD (blue) is predominantly in the middle, but following the switch point, ASE (red) takes over as the middle dendrite for the rest of the length of the bundle. AWA (yellow) is rarely in the middle, and only for short segments (*Figure 2B,C*). This order is maintained throughout larval development, although it becomes increasingly 'noisy' with age. We further confirmed these results by showing that a different set of three amphid dendrites (AWA, YFP; AWC, CFP; ASG, mCherry) are also well-ordered and exhibit a similar switch point (*Figure 2—figure supplement 1C*).

The cause of the switch point is unclear. Similar discontinuities were observed in other bundles by classical EM, and it was noted that 'abrupt changes in neighbourhood exhibited by some neurons' might arise from mechanical obstacles in the local environment or from changes in the composition of the bundle (*White et al., 1986*). Indeed, examination of classical EM sections reveals changes in the overall shape of the bundle along its length, switching from a cylinder to a sheet as it is pressed against the basement membrane of the pharynx closer to the nose (*Figure 2—figure supplement 1D,E*) (*Altun and Hall, 2005*). The composition of the bundle also changes along its length. The neuron AUA extends a dendrite in the posterior portion of the bundle that does not reach the nose, while the amphid socket glial cell extends a process that fasciculates with only the anterior portion

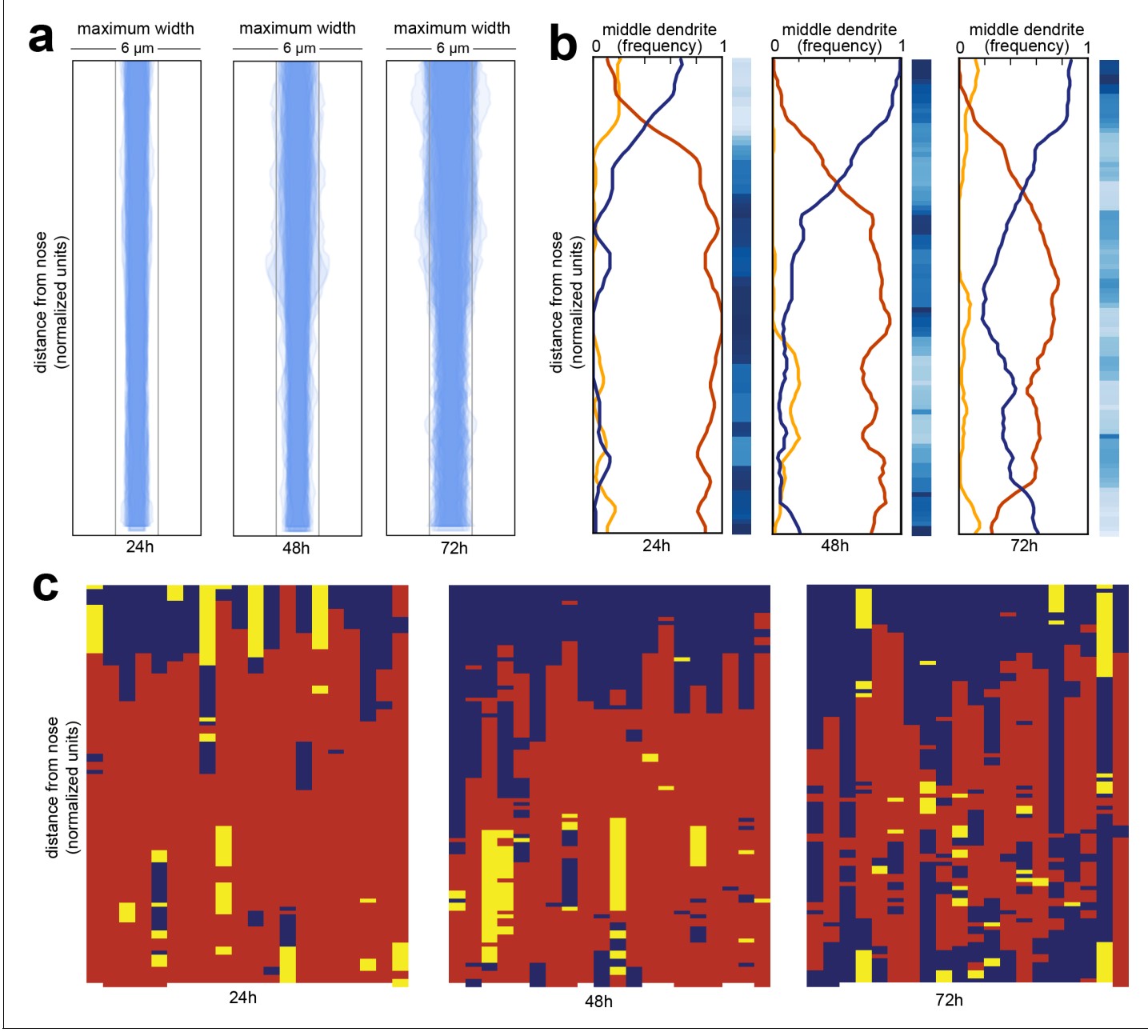

**Figure 2.** Wild-type dendrite bundles are tightly fasciculated and well ordered. (**A**) Maximum distances between AWA, AFD, and ASE dendrites for early stage larvae (24 hr, L2/L3), late stage larvae (48 hr, L4), and young adults (72 hr, one-day adults). Light gray bars, 2 μm width; entire box, 6 μm width. (**B**) Summary plots showing the fraction of animals with AWA (yellow), AFD (blue), or ASE (red) dendrites in the middle at each position along the bundle for 24 hr, 48 hr, and 72 hr time points. Blue color bars show statistical difference from a random distribution (ranked *p*-values, shading as in *Figure 1Cii*); darker shades represent greater difference from random. See *Figure 2—figure supplement 1B* for comparison with chi-squared test *p*-values. (**C**) Population plots showing dendrite 'in the middle' for 24 hr, 48 hr, and 72 hr time points (AWA, yellow; AFD, blue; ASE, red); same data as in (**B**). Each column represents one animal. n = 20 at each time point.

DOI: https://doi.org/10.7554/eLife.35825.003

The following figure supplement is available for figure 2:

**Figure supplement 1.** Dendrite order in wild-type animals.
DOI: https://doi.org/10.7554/eLife.35825.004

of the bundle (*Ward et al., 1975*). We measured the average positions of the AUA dendrite ending and the amphid socket glial cell, and found the position of the switch point does not correlate with the AUA ending, but does correlate with the position where the socket glial cell process joins the bundle (*Figure 2—figure supplement 1F*).

To summarize, we used our imaging pipeline to show that amphid dendrites are well-ordered in young animals and that this order is maintained – albeit imperfectly – during larval growth, despite an approximately two-fold increase in the length and width of the bundle and despite ongoing mechanical perturbations caused by locomotion and pharyngeal pumping of the animal.

## Structures at the dendrite endings are not required for dendrite order

Our data show that amphid dendrites exhibit the most consistent order in the distal region of the dendrite bundle, closest to the nose. This region is especially interesting because it is rich in cell biological structures, including dendritic cilia used to detect signals from the environment and cell-cell junctions between each dendrite and the amphid sheath glial cell (*Figure 3A*). By contrast, the remainder of the dendrites are comparatively featureless, lacking gap junctions, synapses, or any other obvious cell biological specializations. We therefore wanted to test whether cilia or interactions with the sheath glial cell contribute to amphid dendrite order.

We first looked at amphid cilia as a potential source of dendrite order. Early EM reconstructions showed that the order of amphid cilia is also stereotyped, albeit different from that of amphid dendrites (*Ward et al., 1975*). To test the hypothesis that amphid cilia are required for dendrite order, we crossed our markers into a mutant lacking the RFX transcription factor DAF-19, which is required to activate the genetic program for ciliogenesis (*Figure 3B*) (*Swoboda et al., 2000*). Because *daf-19* mutants constitutively enter a non-reproductive developmental state called dauer, we also introduced a mutation in *daf-12*, which encodes a receptor required for dauer entry (*Antebi et al., 2000*). *daf-19; daf-12* animals lack cilia but do not enter dauer, allowing the strain to be maintained as a homozygous stock (*Senti and Swoboda, 2008*). We found that lack of cilia had no effect on fasciculation or dendrite order (*Figure 3B*, *Figure 3—figure supplement 1*). To statistically quantify this observation, we performed permutation tests comparing *daf-19; daf-12* with simulated random distributions (*Figure 3B*, blue column) or with our observed wild-type population (*Figure 3B*, red column). Darker blues indicate that *daf-19* dendrite order is non-random, while the absence of darker reds indicates that *daf-19* dendrite order does not differ from wild type. These tests are consistent with the qualitative impression that dendrite order is unaffected. As cilia are required for normal neuronal activity, this result also implies that dendrite order does not depend on normal patterns of activity.

To further test the role of dendrite endings in establishing dendrite order, we examined mutants lacking the splicing factor MEC-8 (*Lundquist et al., 1996*). Whereas wild-type dendrite endings enter the amphid sheath glial cell in a stereotyped order and converge into a central channel, in *mec-8* mutants the amphid dendrite endings are disorganized and diverge into separate, isolated channels in the sheath glial cell (*Figure 3C*) (*Perkins et al., 1986*). Despite this disorganization, we found that overall dendrite fasciculation and order along the length of the amphid bundle remain normal (*Figure 3C*, *Figure 3—figure supplement 1*). Altogether, our results show that proper arrangement of amphid cilia is not required for dendrite order.

We next tested whether interactions with the amphid glial cell are required for dendrite order. Amphid dendrites form cell junctions with the sheath glial cell, and the sheath glial cell secretes factors that promote the normal development and function of the dendrite sensory endings (*Bacaj et al., 2008*). To test the possibility that the sheath glial cell might impose order on the dendrites, we genetically ablated sheath glia using diphtheria toxin expressed under the control of a late embryonic-stage amphid sheath promoter (*Figure 3D*) (*Bacaj et al., 2008*). We were not able to ablate the sheath glia in earlier embryonic stages, as that causes amphid dendrites to fail to extend (*Singhal and Shaham, 2017*). Our strain also carried a fluorescent marker for the amphid sheath glial cell, allowing us to identify and exclude rare animals in which ablation failed. In glia-ablated animals, we found dendrites to remain well-ordered (blue color bar) and mostly unchanged from wild type (red color bar), suggesting that sheath glia are not required to maintain dendrite order after the amphid bundle initially develops (*Figure 3D*, *Figure 3—figure supplement 1*). Together, our data suggest that the conspicuous cell biological features of amphid dendrites – their cilia and cell-cell junctions – are not required for dendrite fasciculation or order.

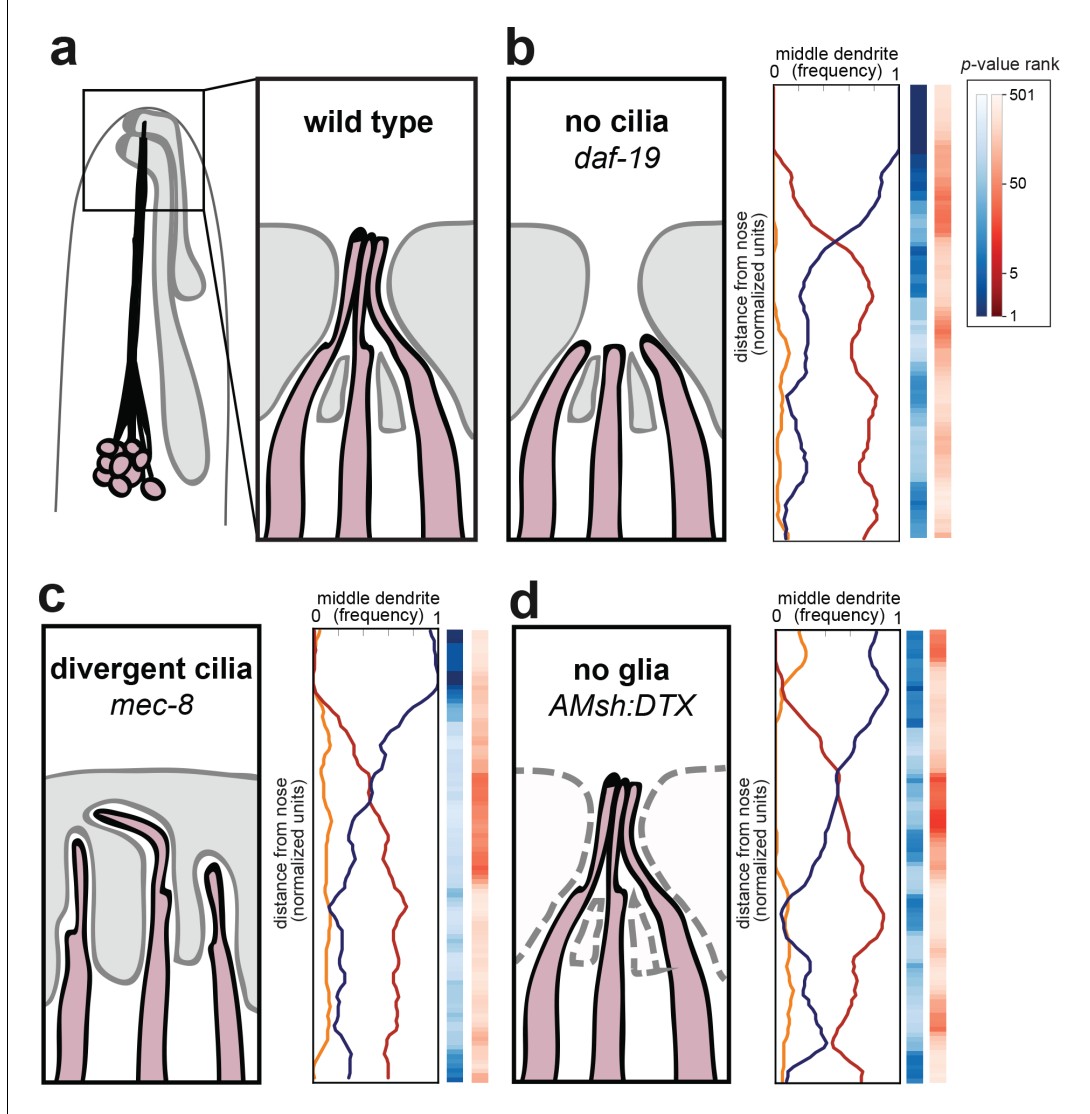

**Figure 3.** Amphid cilia and sheath glia are not required for dendrite order. (**A**) Schematic of wild-type amphid neurons (dark pink) and glial cells (gray). Inset: Amphid dendrites protrude into the sheath glial cell and terminate in cilia. (**B–D**) Dendrite order summary plots for 48 hr (L4) animals with (**B**) no cilia (*daf-19(m86);daf-12(sa204)*, n = 19); (**C**) divergent cilia (*mec-8(u74)*, n = 19) or (**D**) no sheath glia (*AMsh:DTX*, transgene consisting of diphtheria toxin A under control of amphid sheath glial specific promoter, n = 18). Color bars, *p*-value rankings as indicated in key. Blue bars show statistical difference from random (darker shading, less random); red bars show statistical difference from wild type (darker shading, less like wild type).

DOI: https://doi.org/10.7554/eLife.35825.005

The following figure supplement is available for figure 3:

**Figure supplement 1.** Loss of cilia or glia does not disrupt fasciculation.

DOI: https://doi.org/10.7554/eLife.35825.006

## The cell adhesion molecules *cdh-4*, *sax-7*, and *ptp-3* are required for dendrite fasciculation

Next, we considered the possibility that cell adhesion molecules (CAMs) along the lengths of the dendrites might be required for fasciculation and dendrite order. Previous studies in *C. elegans* and other organisms have shown that cell adhesion molecules such as SAX-7/L1CAM and the large extra-cellular matrix molecule DIG-1 are required for dendrite and axon fasciculation (*Burket et al., 2006*; *Bénard et al., 2006*; *Sasakura et al., 2005*). However, few studies in any system have explored how CAMs affect the ordering of axons or dendrites within a bundle (*Lin et al., 1994*; *Schwabe et al., 2014*). We wanted to identify CAMs that affect amphid dendrite fasciculation and order. Since it is

technically unfeasible to conduct a large-scale forward genetic screen for dendrite order defects using our imaging pipeline, we decided to focus first on a small number of candidate CAMs.

To this end, we began by generating a list of 20 CAMs that are known to play roles in axon fasciculation, that have highly enriched expression in amphid neurons, or that interact with CAMs known to affect fasciculation in other areas of the *C. elegans* nervous system or in other organisms (*Table 1*). To further prioritize this list, we reasoned that disrupting CAMs that are important for amphid dendrite fasciculation might cause large-scale dendrite disorganization at some frequency, and that we would be able to detect even infrequent defects by using a dye-filling method to label the amphid neurons. Briefly, animals are soaked in 2 mg/ml of the lipophilic fluorescent dye DiI for 45 min, which for unknown reasons leads to bright and highly specific labeling of six amphid neurons, thus providing a fast and marker-independent method to visualize the overall structure of the amphid bundle. Altered amphid structure can reflect loss of adhesion between dendrites or pleiotropic defects in morphogenesis. We used dye-filling to screen mutations in the 20 candidate CAMs using late larval stage (L4) animals. As a positive control, we found that *dig-1* mutants exhibited readily apparent defasciculation defects in this assay (24% of animals, *Table 1*).

Using this approach, we identified seven additional CAM mutants that cause defasciculation defects (*ptp-3, sax-7, vab-1, sax-3, cdh-4, unc-40* and *nrx-1; Table 1*). Interestingly, even the most pronounced defects among these are weakly penetrant (<25%) suggesting that a redundant system of CAMs contributes to dendrite fasciculation. Loss of *sax-7* was previously observed to cause 'loosening' of amphid dendrites in some animals, along with other defects, but this aspect of its phenotype has not been pursued (*Sasakura et al., 2005*). Because two mutants (*vab-1, sax-3*) exhibited gross head morphology defects and two mutants (*unc-40, nrx-1*) exhibited only very rare defasciculation defects (1 in 50 animals), we chose to focus our initial characterization on the remaining three mutants – *cdh-4, ptp-3*/LAR, and *sax-7*/L1CAM.

## Amphid dendrites are randomly ordered in *cdh-4* mutants

From our candidate screen, we found *cdh-4* to be required for dendrite fasciculation. *cdh-4* encodes a Fat-like cadherin characterized by the large number of cadherin repeats in its extracellular domain (*Figure 4A*). In *C. elegans*, *cdh-4* has been implicated in axon fasciculation in the dorsal and ventral nerve cords (*Schmitz et al., 2007*; *Schmitz et al., 2008*). We used the *rh310* allele, which introduces a premature stop codon in the extracellular domain (*Figure 4A*)(*Schmitz et al., 2008*).

Since only a small number of *cdh-4* mutant animals had dendrite fasciculation defects (6%, *Table 1*), we wanted to ask whether dendrites are still ordered in *cdh-4* individuals that had normally fasciculated dendrite bundles. To do this, we crossed our markers into *cdh-4* animals. Consistent with our dye-filling results, we observed grossly defasciculated amphid dendrites in only 1/21 animals examined (*Figure 4—figure supplement 1A*). Quantification of dendrite distances identified four additional animals in which a portion of the bundle significantly exceeded the average width of wild-type bundles (z-score >3.5, see Methods) and which were therefore also classified as defasciculated. These defasciculated bundles are shown in pink in *Figure 4C* and were excluded from further analysis. In the remaining 16/21 animals, we found that dendrite order was lost despite the bundle remaining intact and well-fasciculated (*Figure 4B*), with the ASE, AFD, and AWA dendrites occupying the middle position with approximately equal frequency along the entire length of the bundle (*Figure 4D*). Permutation tests confirmed that dendrite order in this population is not significantly different from random (light shading in blue color bar) and is significantly different from wild type (dark shading in red color bar).

We found that dendrite order defects are also present in younger animals (24 hr, L2/L3 stage; *Figure 4—figure supplement 1C*). Because amphid dendrites grow by retrograde extension, we considered the possibility that these early defects in dendrite order might reflect mispositioning of neuronal cell bodies relative to each other during embryonic development. However, cell body positioning appeared normal in newly hatched larvae (*Figure 4—figure supplement 1B*).

Our results show that, although *cdh-4* mutants exhibit only partially penetrant defects in fasciculation, they show nearly complete loss of dendrite order within the bundle, suggesting CDH-4 plays an essential role in specifying dendrite order despite having a more redundant role in overall fasciculation.

**Table 1.** Candidate screen to identify factors required for amphid dendrite fasciculation.

Candidate genes were selected from the literature based on known roles in axon fasciculation/guidance, enrichment in amphid neurons ([a]enriched in AWB or AFD, [Colosimo et al., 2004]), or physical interaction with SAX-7, as shown. Animals bearing the designated alleles were subjected to dye-filling, which brightly labels six amphid neurons (AWB, ASH, ASI, ASJ, ASK, ADL), and scored with a fluorescence stereomicroscope for defasciculated amphid bundles. Two mutants exhibited gross head morphology defects concomitant with defasciculation and were not pursued further ([b]gross head morphology defects).

| Candidate gene | Protein information and references | Allele(s) | Animals with defasciculated amphid bundles (percent, n = 50) |
|---|---|---|---|
| Mutants with known axon fasciculation/guidance defects | | | |
| dig-1 | Immunoglobulin (Ig) superfamily; among largest secreted proteins in any animal (~1300 kDa) (**Burket et al., 2006**; **Bénard et al., 2006**) | n1321 | 24 |
| ptp-3 | LAR family protein tyrosine phosphatase (**Ackley et al., 2005**; **Ch'ng et al., 2003**) | mu256 | 14 |
| | | ok244 | 0 |
| sax-7[a] | L1 family/Neuroglian (**Sasakura et al., 2005**; **Wang et al., 2005**; **Zallen et al., 1999**) | ky146 | 12 |
| vab-1 | sole Eph receptor in C. elegans (**George et al., 1998**; **Mohamed and Chin-Sang, 2006**; **Zallen et al., 1999**) | dx31 | 10[b] |
| sax-3 | Robo (Slit receptor) (**Zallen et al., 1998**; **Zallen et al., 1999**) | ky123 | 8 [b] |
| cdh-4 [a] | Fat-like cadherin (**Schmitz et al., 2008**) | rh310 | 6 |
| unc-40 [a] | DCC (Netrin receptor) (**Hedgecock et al., 1990**) | e271 | 2 |
| fmi-1 [a] | Cadherin family (**Najarro et al., 2012**; **Steimel et al., 2010**) | rh308 | 0 |
| dgn-1 [a] | Dystroglycan family (**Johnson et al., 2006**; **Johnson and Kramer, 2012**) | cg121 | 0 |
| syg-1 [a] | IrreC/IRRE family (**Shen and Bargmann, 2003**) | ky652 | 0 |
| casy-1 | Calsyntenin family (**Kim and Emmons, 2017**) | ok739 | 0 |
| Other adhesion molecules enriched in amphid neurons | | | |
| nrx-1 [a] | Neurexin family (**Haklai-Topper et al., 2011**) | wy778 | 2 |
| plx-2 [a] | Plexin (Semaphorin receptor) (**Ikegami et al., 2004**; **Nakao et al., 2007**) | ev773 | 0 |
| nlr-1 [a] | Neurexin/Caspr family (**Haklai-Topper et al., 2011**) | tm2050 | 0 |
| ptp-4 [a] | protein tyrosine phosphatase (**Thompson et al., 2013**) | gk715362 | 0 |
| rig-3 [a] | Ig superfamily (**C. elegans Deletion Mutant Consortium, 2012**) | ok2156 | 0 |
| scd-2 [a] | Receptor tyrosine kinase (**C. elegans Deletion Mutant Consortium, 2012**) | ok565 | 0 |
| igcm-1 [a] | Ig superfamily (**C. elegans Deletion Mutant Consortium, 2012**) | ok711 | 0 |
| Factors that physically interact with SAX-7 | | | |
| dma-1 [a] | Leucine-rich repeat family (**Liu and Shen, 2011**) | wy686 | 0 |

*Table 1 continued on next page*

*Table 1 continued*

| Candidate gene | Protein information and references | Allele(s) | Animals with defasciculated amphid bundles (percent, n = 50) |
|---|---|---|---|
| *mnr-1* | Fam151 family (*Dong et al., 2013*; *Salzberg et al., 2013*) | *wy758* | 0 |

DOI: https://doi.org/10.7554/eLife.35825.007

### Amphid dendrites in *ptp-3*/LAR and *sax-7*/L1CAM mutants exhibit altered order in young animals and random order in adults

We next examined how *ptp-3* and *sax-7* contribute to amphid dendrite fasciculation and order. *ptp-3* encodes a receptor-like protein tyrosine phosphatase that is part of the leukocyte antigen related (LAR) family of proteins that play important roles in nervous system development, including axon guidance and fasciculation (*Ackley et al., 2005*; *Clandinin et al., 2001*; *Dunah et al., 2005*;

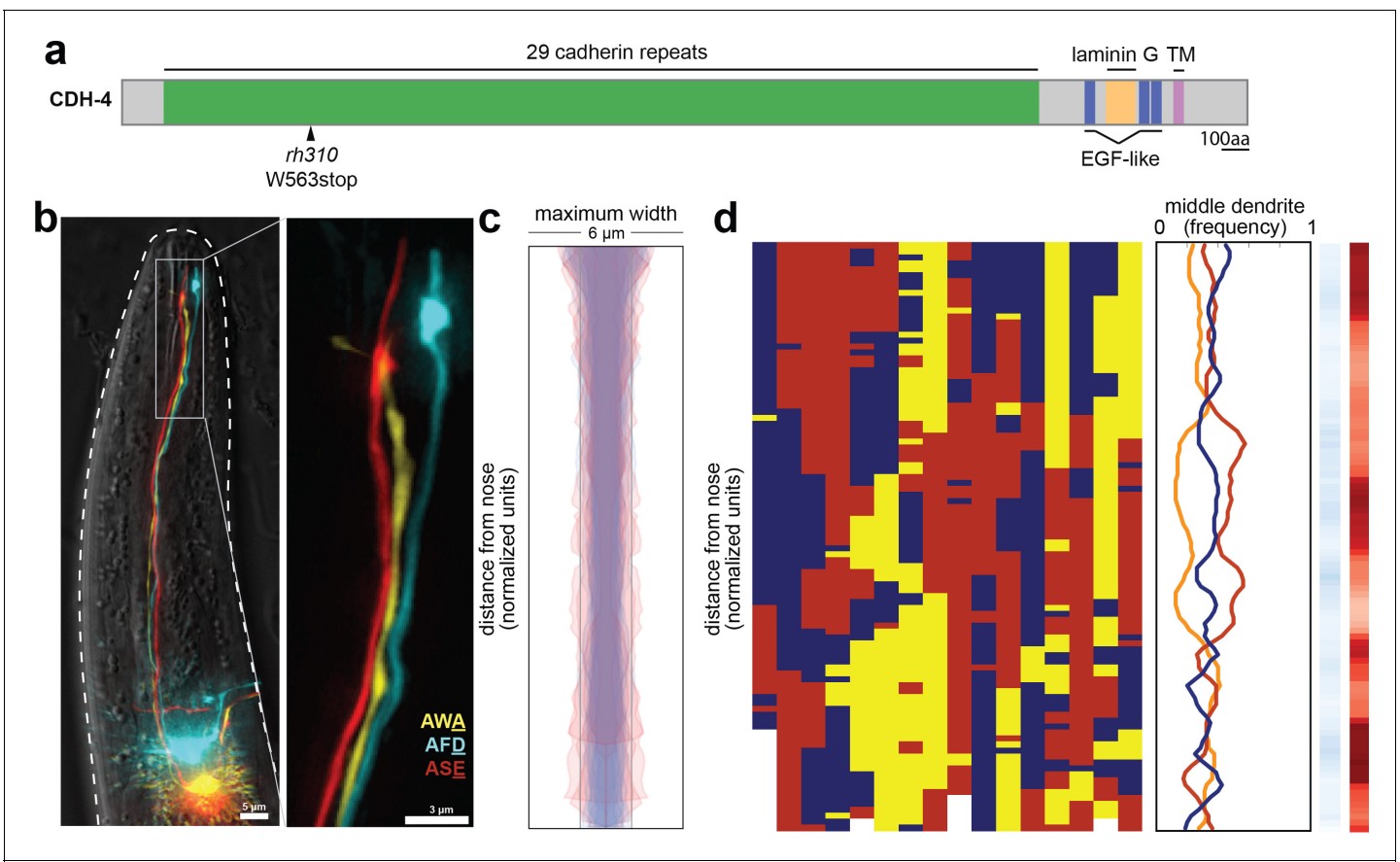

**Figure 4.** *cdh-4* mutants exhibit loss of dendrite order. (A) Schematic CDH-4 protein structure. TM, transmembrane segment; EGF-like, epidermal growth factor-like domain. (B) Maximum-intensity projection of 48 hr (L4) animal showing tightly-fasciculated dendrites with an altered order, compare to *Figure 1A*. (C) Bundle width plots for *cdh-4* (48 hr, L4; n = 21). Five individuals with defasciculated dendrites (see *Figure 4—figure supplement 1A*) are overlaid in pink and were excluded from further analysis. All other animals (n = 16) are overlaid in blue. (D) Population plots and summary plots showing dendrite order for these animals (n = 16). p-value rankings as in *Figure 3*: blue, *cdh-4* vs. random; red, *cdh-4* vs. wild type; darker shading represents greater differences between the populations.

DOI: https://doi.org/10.7554/eLife.35825.008

The following figure supplement is available for figure 4:

**Figure supplement 1.** Fasciculation and dendrite order phenotypes in *cdh-4* mutants.
DOI: https://doi.org/10.7554/eLife.35825.009

*Harrington et al., 2002*; *Krueger et al., 1996*; *Maurel-Zaffran et al., 2001*; *Wills et al., 1999*). *sax-7* encodes a homolog of the L1 cell adhesion molecule that is also widely involved in neurodevelopment, including axon guidance and fasciculation, and is disrupted in a human neurological disorder called L1 syndrome (*Bénard et al., 2012*; *Chen and Zhou, 2010*; *Kim and Emmons, 2017*; *Sakurai, 2012*; *Sasakura et al., 2005*).

To examine dendrite defects, we crossed our markers to *ptp-3* and *sax-7* mutants. There are short and long isoforms of *ptp-3* (*ptp-3a* and *ptp-3b*; *Figure 5A*), and we made use of *ptp-3(mu256)*, which disrupts both isoforms, as well as *ptp-3(ok244)*, which disrupts only the longer *ptp-3a* isoform (*Figure 5A*). There are also short and long isoforms of *sax-7* (*sax-7S* and *sax-7L* respectively (*Bénard et al., 2012*; *Pocock et al., 2008*; *Sasakura et al., 2005*; *Wang et al., 2005*), (*Figure 5B*). We made use of two alleles that disrupt both isoforms – *sax-7(ky146)* was used for most of our analyses, and we confirmed our results using *sax-7(eq1)* (*Wang et al., 2005*; *Zallen et al., 1999*) – as well as *sax-7(nj53)* which prevents expression of *sax-7L* (*Sasakura et al., 2005*).

As expected from our dye-filling assays, *ptp-3(mu256)* and *sax-7(ky146)* exhibit amphid defasciculation defects with low penetrance (pink traces in *Figure 5C,F* and representative images in *Figure 5—figure supplement 1A,D*). Fasciculation defects were not observed in *ptp-3(ok244)*, suggesting that the shorter PTP-3B isoform is sufficient for fasciculation (*Figure 5—figure supplement 1C*). In both *ptp-3(mu256)* and *sax-7(ky146)*, fasciculation defects become progressively more severe throughout larval development (*Figure 5C,F*), consistent with previous observations that SAX-7 is required for the maintenance of nervous system architecture (*Bénard et al., 2012*; *Sasakura et al., 2005*).

To assess dendrite order, we examined only dendrite bundles that remained tightly fasciculated. In older animals (72 hr, adult stage), *ptp-3(mu256)* and *sax-7* exhibited random dendrite order, similar to what we observed in *cdh-4* (*Figure 5D,G*). These defects are specific to the PTP-3B isoform and to SAX-7, as dendrite order is not affected by loss of PTP-3A alone (*ptp-3(ok244)*, *Figure 5—figure supplement 1C*) or by disruption of adhesion molecules that interact genetically or physically with SAX-7 or its homologs in other contexts (DMA-1, CLR-1, IGCM-1) (*Figure 5—figure supplement 2*) (*Dong et al., 2013*; *Islam et al., 2003*; *Liu et al., 2016*; *Salzberg et al., 2013*). While *sax-7* has been reported to exhibit cell body positioning defects in ~10% of newly hatched animals (*Sasakura et al., 2005*), we found that cell bodies remained ordered normally relative to each other in newly hatched animals for both *ptp-3* and *sax-7* (*Figure 5—figure supplement 1B,E*). As a further specificity control, we examined a second allele of *sax-7*, *eq1*, and observed similar defects to *sax-7 (ky146)* (*Figure 5—figure supplement 1F*). Interestingly, *sax-7(nj53)* exhibited defasciculation defects (10/26 animals) but mild or no defects in dendrite order (*Figure 5—figure supplement 1G*).

In contrast to adults, *ptp-3(mu256)* and *sax-7* young larvae (24 hr, L2/L3 stage) exhibited a dendrite order that was neither random nor wild type (*Figure 5D,G*). In *ptp-3(mu256)* animals at the 24 hr (L2/L3) and 48 hr (L4) time points, the bundle is arranged in a stereotyped order that resembles wild type but has the switch point shifted towards the nose, such that ASE (red) occupies the middle position throughout nearly the entire bundle (*Figure 5D*). In *sax-7* animals at the 24 hr (L2/L3) time point, the middle position is occupied with roughly equal frequency by ASE (red) and AWC (yellow) but almost never by AFD (blue), indicating that dendrite order is not random. As animals reach the 48 hr (L4) time point, the order of dendrites appears increasingly random, and by 72 hr (adult) it is indistinguishable from the random order seen in *cdh-4* (*Figure 5G*). Loss of PTP-3 and SAX-7 together (*ptp-3(mu256); sax-7(ky146)* double mutant) leads to even greater defasciculation and increased randomness of dendrite order at the 48 hr (L4) time point (*Figure 5—figure supplement 1H,I*).

Together, our results suggest that PTP-3 and SAX-7 are required, first, to establish the wild-type dendrite order during early development and, second, to maintain dendrite order throughout larval growth. The observation that loss of a single CAM can generate an order that is neither wild type nor random suggests that dendrites might normally 'choose' from among several potential neighbors, and that these preferences are altered when a given CAM is removed.

Recent work has shown that cell-specific differences in the expression level of a single CAM helps to determine neighbor preferences among *Drosophila* retinal axons (*Schwabe et al., 2014*). Therefore, we examined the expression patterns of *ptp-3b* and *sax-7S* by fusing promoter regions upstream of their coding regions (7 kb, *ptp-3b*pro; 3 kb, *sax-7S*pro) to a nuclear-localized mCherry (*Figure 5E,H*). We imaged early larval stage (L1) animals and used the six dye-filling neurons, stained

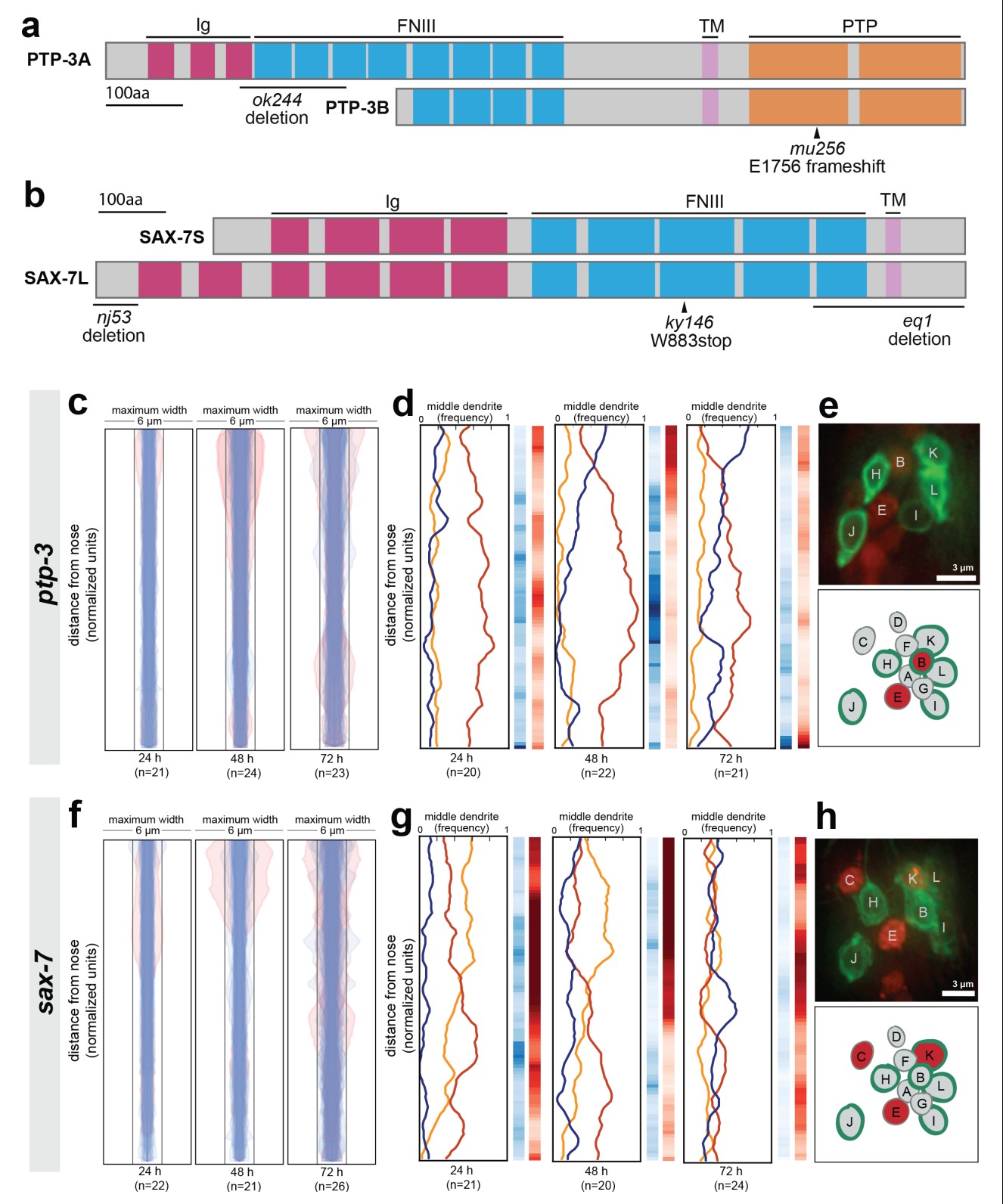

**Figure 5.** Loss of either *ptp-3* or *sax-7* alters dendrite order in young animals. (**A–B**) Schematic protein structures for isoforms of (**A**) PTP-3 and (**B**) SAX-7. TM, transmembrane segment; FNIII, Fibronectin Type III domain; Ig, immunoglobulin-like domain; PTP, protein tyrosine phosphatase domain. (**C, D, F, G**) Bundle width plots and dendrite order summary plots for (**C, D**) *ptp-3(mu256)* and (**F, G**) *sax-7(ky146)* at the developmental stages indicated. Animals with defasciculated dendrites are overlaid in pink on bundle width plots and were excluded from further analysis. (**E, H**) Expression patterns of

*Figure 5 continued on next page*

*Figure 5 continued*

(E) *ptp-3b* promoter and (H) *sax-7S* promoter driving nuclear mCherry (red). Dye-filling (green) was used to label six defined neurons as anatomical landmarks, and remaining neurons were identified by nuclei positions. Maximum-intensity projection images of L1-stage animals, top; schematic showing defined relative positions of amphid nuclei, bottom. A, AWA; B, AWB; C, AWC; D, AFD; E, ASE; F, ADF; G, ASG; H, ASH; I, ASI; J, ASJ; K, ASK; L, ADL.

DOI: https://doi.org/10.7554/eLife.35825.010

The following figure supplements are available for figure 5:

**Figure supplement 1.** Fasciculation and dendrite order phenotypes in *ptp-3, sax-7,* and *ptp-3; sax-7* mutants.

DOI: https://doi.org/10.7554/eLife.35825.011

**Figure supplement 2.** Fasciculation and dendrite order phenotypes in mutants disrupting SAX-7-interacting factors.

DOI: https://doi.org/10.7554/eLife.35825.012

in green, as landmarks to assist in cell identification. We found that *ptp-3b*pro is expressed at low but detectable levels in many cells, including several amphid neurons, with highest expression in AWB and ASE (*Figure 5E*). We found that *sax-7S*pro is expressed throughout the nervous system and, within the amphid, is consistently expressed in AWC, ASE, and ASK (*Figure 5H*). The differential expression we observe using these synthetic reporter constructs is consistent with a model in which these and other CAMs determine dendrite neighbor preferences through differential adhesion, differences in signaling, or both.

To summarize, we found that loss of either *sax-7* or *ptp-3* causes amphid dendrites to initially take on a non-random arrangement that is distinct from wild type. In both *sax-7* and *ptp-3* mutants, this order becomes increasingly random over time, and correlates with increased defasciculation. We also found that *sax-7* and *ptp-3* were expressed in a subset of amphid neurons, suggesting that these adhesion molecules are not uniformly expressed. Taken together, our data suggest that *sax-7* and *ptp-3* are required for ordering dendrites within the amphid bundle, such that loss of either CAM causes a weakly-penetrant defasciculation phenotype and a highly-penetrant change in dendrite order.

## Misexpression of SAX-7 alters amphid dendrite order

We considered two models to explain how CAMs promote dendrite order. In the first model, dendrite order is determined during development through a CAM-independent mechanism, for example cell lineage. CAMs would then act like concrete poured over this pre-existing structure to secure it in place such that, without appropriate CAMs, dendrite order would be labile and deteriorate over time. In the second model, dendrite order is determined by differential expression of the CAMs themselves acting as adhesion molecules, signaling receptors, or both. In this model, experimentally misexpressing a single CAM might alter dendrite order (see concept model, *Figure 6A*).

To test this idea, we misexpressed SAX-7 using promoters that are expressed in one, several, or many additional amphid neurons (*odr-10*pro, *ptp-3*pro, *osm-6*pro respectively). We used SAX-7 because, in our hands, PTP-3 appears to undergo post-transcriptional regulation that makes it difficult to manipulate. Misexpression of SAX-7 did not change the dendrite order drastically, but misexpression in more cells and at higher levels led to subtle yet reproducible changes in the wild-type order, shifting the switch point more posteriorly (*Figure 6D*). These results do not exclude that additional CAM-independent mechanisms might contribute to dendrite order, but suggest that SAX-7 may play an instructive role in specifying the arrangement of amphid dendrites, possibly by contributing to patterns of differential adhesion.

## Discussion

### Principles of nerve bundle organization

In this study, we took advantage of the simple nervous system of *C. elegans* to quantitatively assess the arrangement of individual dendrites within a nerve bundle. We found that amphid dendrites are ordered within the bundle, and this order is maintained over time. Dendrite order does not seem to depend on the conspicuous cell biological structures at the dendrite endings (sensory cilia and dendrite-glia junctions) but is instead imposed by multiple CAMs expressed by the neurons. Loss of the

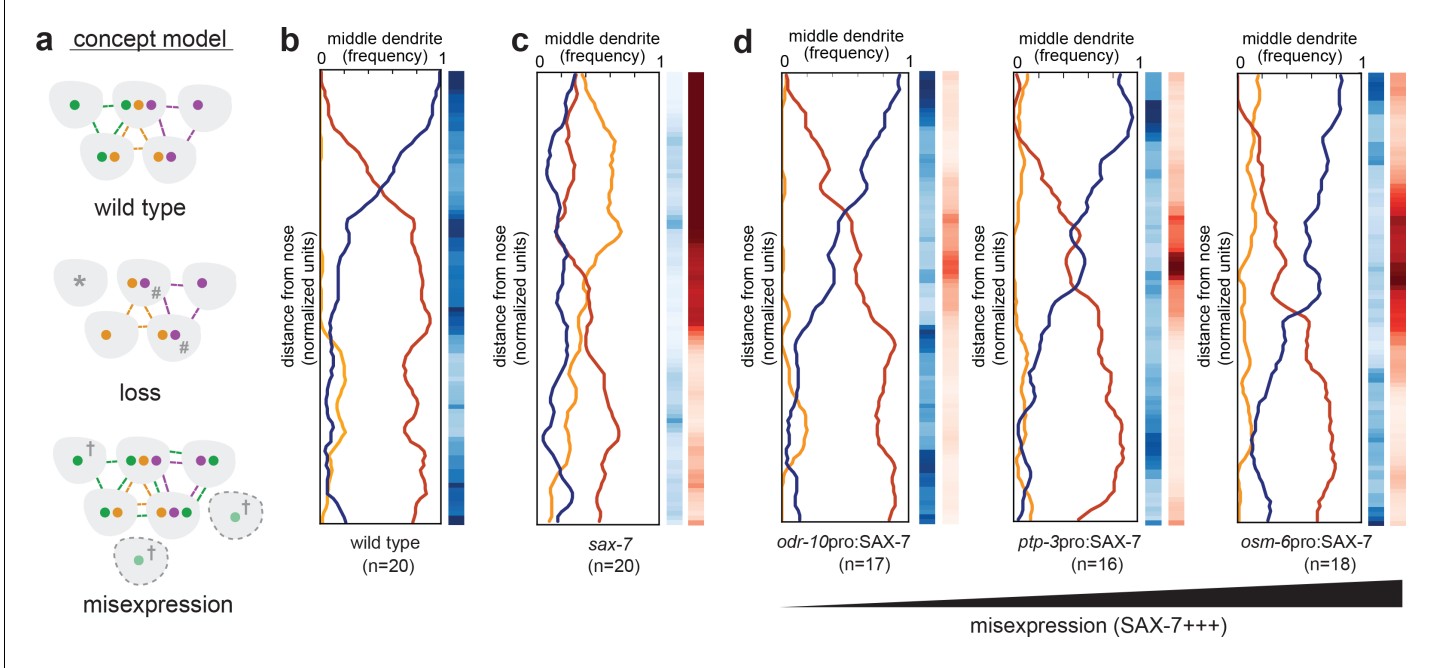

**Figure 6.** Misexpression of SAX-7 in all amphid neurons alters dendrite order. (A) Concept model illustrating how differential expression of adhesion molecules (green, orange, purple) might lead to a stereotyped arrangement of dendrites (gray). In this version, loss of the green adhesion molecule might cause defasciculated dendrites (*) and more variable order due to dendrites becoming equivalent (#). Misexpression of the green adhesion molecule might lead to more variable order or to a different order due to altered partner preferences (†). (B–D) Dendrite order summary plots and statistical tests comparing (B) wild type (same data as *Figure 2B*), (C) *sax-7(ky46)* (same data as *Figure 5G*), and (D) misexpression of SAX-7S cDNA (SAX-7+++) under control of the indicated promoters: *odr-10*pro, strong expression in a single amphid neuron (AWA); *ptp-3*pro, low-level expression in several amphid neurons plus strong expression in ASE and AWB (see *Figure 5E*); *osm-6*pro, strong expression in most amphid neurons (see *Figure 5E, H*). 48 hr (L4) animals. *p*-value rankings as in *Figure 3*: blue, indicated strain vs. random; red, indicated strain vs. wild type; darker shading represents greater differences between the populations.

DOI: https://doi.org/10.7554/eLife.35825.013

Fat-like cadherin CDH-4 leads to randomized arrangement of dendrites, whereas loss of PTP-3/LAR or SAX-7/L1CAM, or misexpression of SAX-7, can cause dendrites to take on an altered non-random order. Taken together, our results suggest that the arrangement of dendrites within the bundle is determined by multiple CAMs, and altering their relative expression can lead to novel arrangements.

Our results provide insight into the organization of nerve bundles, a defining feature of the nervous system that has previously received relatively little attention due to the technical challenges of studying it. Bundles are known to sometimes correspond to functional groupings of axons, for example each of the cranial nerves is composed of axons with shared functions such as smell, vision, or oculomotor control. Similarly, in the periphery, 'fast' and 'slow' motor axons are segregated into distinct bundles during development (*Milner et al., 1998*). In the mammalian cortex, dendrites arising from neurons in different cortical layers come together to form microcolumns (*Fleischhauer et al., 1972*; *Fleischhauer, 1974*; *Peters and Walsh, 1972*). Axons projecting from the retina, as well as the olfactory bulb, have provided evidence that neurites can exhibit stereotyped arrangements within their respective bundles, for example reflecting their topographic or dorsal-ventral origins, or the chronological order of their outgrowth (*Chan and Chung, 1999*; *Walsh and Guillery, 1984*). Recent studies have shown that these arrangements arise during development and can affect axon targeting (*Cioni et al., 2018*; *Imai et al., 2009*; *Zhou et al., 2013*). However, the question of how nerve bundles are organized has been difficult to study due to two major technical problems. First, fasciculation defects often lead to axon guidance defects, and vice versa, making it challenging to distinguish one from the other except in rare cases (*Luxey et al., 2013*). Our approach circumvents this problem by taking advantage of the fact that amphid dendrites grow collectively via retrograde

extension (*Heiman and Shaham, 2009*). Second, most studies have only been able to assess large populations of axons, making it impossible to ask how single neurons or neuron classes are arranged within a bundle. We and others have overcome the latter limitation by turning to invertebrate model organisms with simpler and more stereotyped anatomy that allows the trajectories of defined neurons to be examined.

Two previous sets of studies using invertebrate models are especially helpful to compare with our results. In classical studies using grasshoppers, Goodman and colleagues showed that a certain growing (G) axon selectively adheres to a defined partner (P) axon in a specific bundle. If the P axon is ablated, the G axon does not adhere to other axons in this bundle or in any neighboring bundles, suggesting a remarkable degree of selective adhesion between G and P (*Bastiani et al., 1984*; *Lin et al., 1994*; *Raper et al., 1984*). In a more recent study, Schwabe *et al.* studied the cartridges of the *Drosophila* visual system, which exhibit an invariant organization in which neurites from lamina cells (L) are surrounded by photoreceptor (R) axons. They found that this organization reflects differential expression levels of N-cadherin, with the higher-expressing L neurites forming an adhesive core surrounded by the lower-expressing R axons (*Schwabe et al., 2014*).

These studies led to two major principles, which are further supported by the work described here: first, that neurites can exhibit stereotyped neighbor relationships within a bundle and, second, that these relationships can arise from differential adhesion. Importantly, our study introduces a third principle – that multiple adhesion molecules can act in concert to specify distinct relationships among many neurites, in a pattern that is more complex than the 1:1 pairing of G and P axons or the inside:outside relationship of L and R neurites.

## Functional consequences of ordered bundles

Why might nerve bundles be ordered? One trivial explanation would be that bundle order reflects the developmental order of neurite outgrowth, with the oldest neurite – typically the pioneer axon or dendrite – in the center, and the youngest neurites on the periphery. However, this is not the case in *C. elegans*, the only organism in which the birth order and neurite position of every neuron is known. As an example, the VD motor neurons are born post-embryonically and extend axons along the ventral nerve cord after the nerve cord is established, yet their axons insinuate deeply into the ventral nerve cord to adhere closely to the DD motor neurons, a functionally similar set of motoneurons that are born much earlier and lie at the center of the ventral nerve cord (*White et al., 1976*; *White et al., 1986*). This and similar examples suggest that neurites 'choose' their neighbors within a bundle, possibly with functional consequences.

One intriguing hypothesis is that one's immediate neighbors within a bundle influence neuronal activity. For example, the dendrites of gonadotropin-releasing hormone neurons are intertwined in bundles, and this arrangement has been proposed to help synchronize hormone release (*Campbell et al., 2009*). In a possibly related example, neuropeptide-releasing neurons in *Drosophila* that serve as circadian pacemakers undergo daily changes in axon fasciculation, driven by cycling expression levels of the NCAM adhesion molecule FAS2 (*Fernández et al., 2008*; *Sivachenko et al., 2013*). Interestingly, genetically manipulating FAS2 expression to force constitutive defasciculation leads to changes in circadian behavior, suggesting that fasciculation may indeed affect neuronal activity (*Sivachenko et al., 2013*).

There are at least three ways in which neighboring neurites could influence each other's activity. First, the presence of chemical synapses or electrical synapses (gap junctions) would couple the activity of immediate neighbors. Notably, no such synapses are present along the amphid dendrites. Second, neuropeptides or other small secreted molecules could locally signal to one's immediate neighbors. Finally, neighboring axons or dendrites could affect each other through passive electrical properties, a concept referred to as ephaptic coupling (*Anastassiou et al., 2011*; *Arvanitaki, 1942*). This idea is exciting because it is highly generalizable – anywhere neurites are apposed to one another, such ephaptic effects could be manifested. Consistent with this notion, Ferenczi *et al.* used an engineering approach to optogenetically manipulate ion flow in hippocampal and cortical axons and observed a 'bystander' effect on the membrane current of neighboring axons, independent of synaptic transmission (*Ferenczi et al., 2016*). Remarkably, in wild-type *Drosophila*, Carlson and colleagues demonstrated similar ephaptic effects between bundled olfactory dendrites (*Su et al., 2012*): a short pulse of odor-induced activity in one of these dendrites inhibited tonic firing of its

bundled neighbor, independently of chemical or electrical synapses. Importantly, they showed that this ephaptic effect could even alter behavioral responses.

In light of these observations of ephaptic coupling, our finding that dendrites assume stereotyped neighbor relationships could have important implications for how sensory information is processed. Recently developed methods for whole-brain calcium imaging in *C. elegans* make it an ideal system for asking whether the neighbor relationships we describe here are mirrored by patterns of correlated neuronal activity and, if so, whether such correlations are disrupted by the mutants we found to alter dendrite order (*Nguyen et al., 2016*; *Prevedel et al., 2014*; *Schrödel et al., 2013*).

## How simple rules generate complex patterns

As Warren Lewis, a pioneer in cell biology, put it in 1922: 'Were the various types of cells to lose their stickiness for one another and for the supporting extracellular white fibers, reticuli, etc., our bodies would at once disintegrate and flow off into the ground in a mixed stream of ectodermal, muscle, mesenchyme, endothelial, liver, pancreatic, and many other types of cells' (*Lewis, 1922*). Yet, it is not enough just to stick. Cells care about their neighbors, and arrange themselves into elaborate patterns as they assemble complex organs like the brain. Although it is important to note that CAMs also play major roles in signaling, our work points to the possibility that differential adhesion between cells might contribute to the emergence of biological order. In this model, the arrangement of cells in a tissue is optimized for adhesive strength between them – cells would thermodynamically 'fold' into their final arrangement in a manner comparable to protein folding. Importantly, if a single adhesion molecule is absent or misexpressed, the system would reorder itself to maximize adhesion, thus producing a new stable order.

Evolutionarily, this 'optimize adhesion' rule provides a general strategy to create a diverse set of well-ordered structures using only small changes in adhesion molecule expression. Interestingly, amphid sensilla in other nematode species such as *Acrobeles complexus*, *Strongyloides stercoralis*, and *Haemonchus contortus* also exhibit well-ordered bundles, albeit with configurations that are different from each other and from *C. elegans* (*Bumbarger et al., 2009*). It is interesting to speculate whether this diversity of structures may have arisen partly through altered expression of a small number of CAMs.

# Materials and methods

## Strains and maintenance

Strains were constructed in the N2 background and cultured under standard conditions (*Brenner, 1974*; *Stiernagle, 2006*). In addition to the wild-type strain N2, the transgenes and strains used in this study are described in *Supplementary files 1–3*. Unless otherwise specified, all animals were imaged in the L4 stage, corresponding with ~48 hr after bleach synchronization (see 'Time point analyses' below).

## *cdh-4* strain construction

To examine dendrite order in *cdh-4* mutant animals, we crossed *cdh-4(rh310)* animals into a strain expressing a three-neuron marker (CHB2646). Because *cdh-4* is located on the same chromosome as one of our integrated fluorescent markers (*hmnIs23*, AWA:YFP), we created a strain (CHB2770) carrying AWA:YFP on an extrachromosomal array (*hmnEx1486*) and crossed that array into a strain containing *hmnIs17* (AFD:CFP, ASE:mCherry) for all *cdh-4* analyses. For the permutation test comparing *cdh-4* to wild-type dendrite order, we used CHB2646 instead of CHB1963 as the wild-type control.

## Time point analyses

Animals were bleach-synchronized (20% bleach, 250 mM NaOH in dH$_2$O for 5 min, then hatched overnight in M9 medium), plated onto agar plates containing food, and cultured under standard conditions. We imaged animals at three different time points: second larval stage (L2 stage; 24 hr after plating), fourth larval stage (L4 stage; 48 hr after plating), and 1 day adult (72 hr after plating).

## Microscopy

Image stacks were collected on a DeltaVision Core imaging system (Applied Precision) with a UApo 40×/1.35 NA oil-immersion objective (72 hr animals) or a PlanApo 60×/1.42 NA oil-immersion objective (24 hr and 48 hr animals) and a Photometrics CoolSnap HQ2 camera (Roper Scientific). Animals were mounted on an agarose pad with 20–40 mM sodium azide and imaged in yellow (excitation [EX] 513 nm/emission [EM] 559 nm), red (EX 575 nm/EM 632 nm), blue (EX 438 nm/EM 470 nm), and/or green (EX 475 nm/EM 525 nm) channels. To avoid possible complications due to left-right asymmetry, animals were selected for dendrite bundle imaging such that their right-hand side faced the coverslip and only this bundle was imaged.

Deconvolution and analysis of images were performed with Softworx (Applied Precision) and ImageJ (NIH, Bethesda, MD). Maximum-intensity projections were obtained using contiguous optical sections.

## Notes on image processing

Projections were adjusted for brightness, contrast, and were pseudo-colored in Photoshop (Adobe). Merged color images were assembled using the Screen layer mode in Photoshop. Figures were assembled using Photoshop CS5.1 and Illustrator CS5.1.

## Image analysis

Image analysis for each animal was done in three parts. First, we imaged three amphid neurons using our three-neuron marker (ASE:mCherry, AWA:YFP, AFD:CFP). We then input this image stack into a custom-made Matlab script that detected the 3D coordinates of each dendrite and generated files containing the distances between each pair of dendrites (AWA-ASE, AWA-AFD, and AFD-ASE) at every point along the dendrite bundle (see Section I: generating 3D coordinates and pairwise distances). Second, we manually inspected and corrected the computer-generated dendrite traces (see Section II: manual inspection and selection of dendrite traces). Finally, we pooled animals belonging to the same population and wrote scripts in Python to generate figures to visualize the data as well as conduct statistical tests to compare populations (see Section III: data visualization, resampling methods, and statistical analysis). Scripts for all the analysis in this paper are available for download at http://github.com/zcandiceyip/dendrite_fasciculation (**Yip, 2018**; copy archived at https://github.com/elifesciences/dendrite_fasciculation).

### Section I: Generating 3d coordinates and pairwise distances

The first step in our image-processing pipeline was to obtain 3D coordinates for the ASE, AWA, and AFD dendrites in each animal. To do this, we wrote a script in Matlab that takes an image stack containing three amphid dendrites imaged in three different channels as input and returns several files. One file contains pairwise distances between the three dendrites along the length of the dendrite bundle starting at the dendrite tip. Another set of files contains the digitized dendrite traces superimposed on maximum-intensity projections for manual inspection of the trace accuracy. First, we will describe how the pairwise distances between dendrites are generated.

As mentioned above, our script takes as input an image stack containing three amphid neurons imaged in three different channels. For each amphid neuron, we manually selected the start and end points of the dendrite by clicking on 2D projections of that image generated in Matlab. We consistently chose the starting point to be the ciliated ending at the tip of the dendrite and the ending point to be the cell body. Next, for each dendrite, Dijkstra's algorithm was used to find the brightest path between the start and end points. Dijkstra's algorithm finds the shortest path between two nodes, where the objective function is to minimize the distance between the two nodes. In this case, the two nodes are the user-defined start and end points, and distance is defined by the inverse of the intensity of each pixel between the start and end points. Thus, the brightest pixels between the start and end points represent the shortest path. Since the brightest pixels in our image stack between the start and end points correspond to the dendrite itself, Dijkstra's algorithm yields the 3D coordinates of the dendrite, thus generating a computerized trace of that dendrite. As described below ('Manual inspection and selection of dendrite traces'), dendrite traces were cropped at a later step to exclude the distal dendrite ending with its complex cilium as well as the proximal portion that crosses the nerve ring (see **Figure 1A**, arrowheads).

After applying Dijkstra's algorithm to obtain digital traces of the three amphid dendrites, the next step is to generate a centroid line that runs in the middle of the three dendrites. To do this, we first cropped the start and end points of the three dendrites so that the three dendrite traces were of similar lengths. We did this by averaging the coordinates of the three dendrite tips on each end, selecting the dendrite that gave the shortest distance between the average dendrite tip and the opposite end of that dendrite, and finding dendrite points on the other two dendrites that were closest to the starting point of the shortest dendrite. Those two dendrite points became the new starting points for the two longer dendrites. This process was repeated for the other side of the dendrite bundle. Next, we generated new start and end points of the dendrite bundle by averaging the coordinates of the three cropped dendrites on each end, and found a centroid line by averaging the coordinates of all the points along the three dendrite traces. Finally, we used that centroid line to determine a series of planes that intersect each of the three dendrite traces once by walking along the average trace pixel by pixel, and used Dijkstra's algorithm to find the shortest path between the centroid line to each of the three traces.

These planes, or cross-sections, yield triangles where each vertex of the triangle is a point along one of the three dendrite traces and the sides of the triangle give the pairwise distances between amphid dendrites in a single cross-section. We then calculated each of these pairwise distances by finding the Euclidean distance between two vertices, and saved them in a comma-delimited ('.csv') spreadsheet. In addition, we obtained 2D projections of the computerized traces of each dendrite and superimposed them onto maximum-intensity projections of each image stack to manually confirm that the dendrite trace followed the actual dendrite in the image.

## Section II: Manual inspection and selection of dendrite traces

To confirm that the computer-generated dendrite traces and resulting pairwise distances for each animal were accurate, we manually inspected maximum-intensity projections of each image stack projected in XY- and YZ-planes and superimposed the computer-generated dendrite trace onto the projections. If the computer-generated dendrite trace did not follow the dendrite in the maximum-intensity projections, we excluded that animal in further analyses.

In general, the computer-generated dendrite traces followed the dendrites with high fidelity. However, the start and end points of the computer-generated traces were usually inaccurate, as the chosen start and end points were selected a few microns outside the actual start and end points of the dendrite. To correct these inaccuracies and standardize the start and end points, we used the XY-projections to count the number of pixels that were traced inaccurately at the proximal and distal dendrite ends, and deleted that number of rows in the corresponding '.csv' spreadsheet. This technique ensured that we were looking at dendrite tracings that, across all animals, begin at the dendrite tip (excluding the cilium) and end near the nerve ring (see *Figure 1A*).

To correct for variations in animal size within a population, we measured the length of an anatomical feature of the head of each animal (distance from the anterior bulb of the pharynx to the nose tip) and used that distance to normalize the lengths of the dendrite traces. Because we are making point-by-point comparisons along the dendrite bundle across animals, we further segmented each dendrite bundle into 100 equally-sized bins and took the mean pairwise distance within each bin for further analysis. For example, if an animal had pairwise distance measurements for 200 positions along the dendrite bundle, then we would segment the bundle into 100 bins, with each bin containing pairwise distance measurements from two adjacent positions, and we would assign the mean of those two pairwise distance measurements to that bin. The purpose of binning is to normalize dendrite bundle positions relative to head size; thus, in each individual, bin 50 is at the same position relative to anatomical landmarks but at a different physical distance from the nose (in μm) depending on head size.

## Section III: Data visualization, resampling methods, and statistical analysis
### Maximum pairwise distance plots and defasciculation

For each animal, we plotted the maximum pairwise distance (bundle width) at each position along the dendrite bundle. We defined a dendrite bundle to be defasciculated by comparing its bundle width at each position to the distribution of bundle widths observed at that position in age-matched wild-type animals. If the bundle width exceeded the wild-type average by more than 3.5 standard

deviations for at least 10 consecutive positions, then that bundle was considered defasciculated. For reference, in a normally distributed dataset, approximately 1 in 2000 data points exceeds the mean by at least 3.5 standard deviations, and our bundle width dataset consists of 526 dendrite bundles with their widths measured at 100 positions each (52,600 data points). Independently, each bundle was also inspected visually and subjectively classified as fasciculated or defaciculated; these manual calls differed from the automated classification in <10% of cases (46 of 526 bundles). Animals with defasciculated dendrite bundles are overlaid in pink on the bundle width plots and were excluded from further dendrite order analysis.

## Population plots and summary plots

Each column in the population plot represents the middle dendrite (ASE in red, AWA in yellow, AFD in blue) at each of the 100 bins of the bundle of a single animal. To create summary plots, we calculated and plotted three fractions – the counts of ASE, AFD, and AWA as a fraction of the total for each bin along the dendrite bundle. For populations with ordered dendrite bundles, one of those three fractions should be close to 1 while the other two fractions should be close to 0, whereas populations with highly disordered bundles have all three fractions closer to 0.33.

## Statistical analyses

We used two approaches to test whether the dendrite order for a population is significantly different from random. First, we used a chi-squared test to test for the independence of two populations (genotype vs. random). We chose the chi-squared test because our data is categorical (at each position, the middle dendrite is either AWA, ASE, or AFD) and, for our typical sample sizes (~20), counts of >5 are expected in each category for a random distribution. We calculated the chi-squared values (chi-squared test statistic: $\sum \frac{(observed-expected)^2}{expected}$) and associated $p$-values at each point along the dendrite bundle, where the observed are the middle dendrite counts of ASE, AWA, and AFD for a given genotype and the expected values are $n/3$, where $n$ is the number of dendrite bundles analyzed for the given genotype. For example, if n = 21 then we would expect a random distribution to yield 7 counts each of ASE in the middle, AWA in the middle, and AFD in the middle, and these would be compared to our observed values using the formula above.

When comparing a mutant genotype to wild type, we could not use the chi-squared test because the expected value at some positions would be <5 (for example, at the nose tip in 48h L4 animals, both ASE and AWA values are zero, leading to a division by zero error when attempting to use the chi-squared formula; the chi-squared test is also not suitable for comparisons where expected values are <5 in any category). Therefore, we adopted a second approach based on a permutation test using Fisher's exact test (3 × 2) as the test statistic. In this case, the null hypothesis is that the two populations (for example, mutant genotype vs. wild type) are drawn from the same distribution and differ only by sampling error. We first calculated a nominal $p$-value at each position along the bundle by comparing the counts of AWA, ASE, and AFD of each population using Fisher's exact test. Then, for 500 iterations, we merged the two populations of middle dendrite values, randomly split the mixed populations into two equally-sized groups, and calculated $p$-values comparing these resampled mock populations to each other using Fisher's exact test. This approach yields 501 $p$-values (1 true p-value+500 $p$-values from resampling) for each point along the length of the dendrite bundle. Finally, we determined the percentile rank of the true $p$-value and plotted that rank using a red log-scale color bar. Darker reds on the color bar indicate that the true $p$-value is much lower than would be expected if the samples were drawn from the same population; that is, the populations are more different than one would expect from sampling error alone.

We also used this approach to compare each genotype to a random population (*Figures 2–6*, *Figure 2—figure supplement 1*, *Figure 4—figure supplement 1*, *Figure 5—figure supplements 1–2* blue bars). In all figures, rather than presenting a mixture of chi-squared and permutation tests, permutation test results are shown for comparison to random (*Figures 2–6*, *Figure 2—figure supplement 1*, *Figure 4—figure supplement 1*, *Figure 5—figure supplements 1–2* blue bars) and comparison to wild type (*Figures 3–6*, *Figure 4—figure supplement 1*, *Figure 5—figure supplements 1–2*, red bars). A comparison of chi-squared and permutation tests is shown in *Figure 2—figure supplement 1B* (green vs. blue bars respectively).

## Switch point swarmplot

To create this swarmplot (*Figure 2—figure supplement 1F*), we measured three different lengths in wild-type L4 animals. First, we measured the distance from the switch point to the nose tip. To do this, we defined the switch point to be the point where the middle dendrite changes from one dendrite to another for more than one bin (for definition of a bin, see Section II under 'Image analysis' methods section). Second, we measured the distance from the nose tip to the dendrite tip of AUA. Finally, we measured the length of the amphid socket that is fasciculated with the amphid; we used the amphid dendrite AFD (blue) as a proxy for the amphid bundle. We created the figure using the swarmplot function in the Python Seaborn package. To determine the difference between populations, we used a two-sample Kolmogorov-Smirnov test because the measurements for the switch point were not normally distributed.

## Candidate screen of cell adhesion molecules

For each mutant, we used DiO (Sigma, D4292) to dye-fill six amphid neurons and scored 50 L4 animals for amphid defasciculation phenotypes using a fluorescence dissecting microscope (*Altun et al., 2002*).

## Expression pattern analysis

To determine the identity of the amphid neurons that express *sax-7* or *ptp-3* we used DiI (Sigma, 468495) to dye-fill L1 animals expressing a nuclear-localized mCherry (NLS-mCherry-NLS) under the control of *sax-7* or *ptp-3* promoters (CHB1687 and CHB1840 respectively, see Supp. File 1). We then collected image stacks for these animals in the red and green channels as L1-stage animals as well as positions of all nuclei, visualized using Nomarski optics. Because the identity and positions of the dye-filled amphid neuron cell bodies are known for L1 animals (*Sulston et al., 1983*), we used the position of the dye-filled amphid cell bodies to infer the identity of the other amphid neurons based on their relative nuclei positioning.

## Acknowledgements

We thank Tiao Xie and Hunter Elliott of the Harvard Image and Data Analysis Core for writing code for image analysis, and Josh Sanes, Elizabeth Engle, and members of the Heiman lab for comments on the manuscript. We thank Hiroyuki Sasakura and Ikue Mori for the SAX-7S cDNA. We thank the CGC, which is funded by NIH Office of Research Infrastructure Programs (P40 OD010440), and WormBase. We thank David Hall for permission to use Slideable Worm images. This work was supported in part by an NSF Graduate Research Fellowship and Alfred J Ryan Foundation Fellowship (ZCY) and NIH R01GM108754 and the Harvard Milton Fund (MGH). The authors declare that they have no competing interests.

## Additional information

### Funding

| Funder | Grant reference number | Author |
|---|---|---|
| National Institutes of Health | R01GM108754 | Maxwell G Heiman |
| National Science Foundation | Graduate Student Research Fellowship | Zhiqi Candice Yip |
| Harvard University | Milton Fund | Maxwell G Heiman |

The funders had no role in study design, data collection and interpretation, or the decision to submit the work for publication.

### Author contributions

Zhiqi Candice Yip, Investigation; Maxwell G Heiman, Supervision

**Author ORCIDs**
Maxwell G Heiman http://orcid.org/0000-0002-2557-6490

**Decision letter and Author response**
Decision letter https://doi.org/10.7554/eLife.35825.019
Author response https://doi.org/10.7554/eLife.35825.020

## Additional files

**Supplementary files**
• Supplementary file 1. Strains used in this study
DOI: https://doi.org/10.7554/eLife.35825.014

• Supplementary file 2. Transgenes used in this study
DOI: https://doi.org/10.7554/eLife.35825.015

• Supplementary file 3. Plasmids used in this study
DOI: https://doi.org/10.7554/eLife.35825.016

• Transparent reporting form
DOI: https://doi.org/10.7554/eLife.35825.017

**Data availability**
A browser (http://heimanlab.com/ibb) has been developed to provide access to the extensive underlying dataset (475 fasciculated dendrite bundles consisting of three pairwise distance measurements and corresponding p-value rankings at 100 positions per bundle). Code used for data analysis is available at http://github.com/zcandiceyip/dendrite_fasciculation (copy archived at https://github.com/elifesciences/dendrite_fasciculation).

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
