## [Decision Letter]

Thank you for submitting your article "Ordered arrangement of dendrites within a *C. elegans* sensory nerve bundle" for consideration by *eLife*. Your article has been reviewed by Didier Stainier as the Senior Editor, a Reviewing Editor, and two reviewers. The reviewers have opted to remain anonymous. The reviewers have discussed the reviews with one another and the Reviewing Editor has drafted this decision to help you prepare a revised submission.

Generally, the overall relevance of the problem addressed is much appreciated by the reviewers, so is the quality of the data and the potential interest of the conclusions. However, there are a number of issues raised by the reviewers (listed below in the individual reviewers’ comments) that both reviewers and the editor feel are very important to address. The issues that particularly stand out are the questions relating (a) to a developmental or a maintenance role of the studied adhesion molecules and (b) the specificity of *sax-7* function (is it a generic glue or does it have more specific, instructive roles). These issues shall not simply be addressed by changes in the text but require some very straight-forward experimentation, as suggested by the reviewers.

Reviewer #1:

In this paper Yip and Heiman address how dendrites are organized within nerves. Understanding of how nervous systems assemble and how nerves are organized in bundles is an important problem, which remains poorly understood. Through a simple, but clever method they use three dendritic processes in the nose of *C. elegans* as a proxy to describe a stereotypic organization of the amphid sensory bundle, which consists of a total of 11 dendrites in *C. elegans*. The authors show that the stereotypic organization of bundled amphid dendrites is independent of their ciliated endings or associated glia. A candidate genetic screen using cell adhesion molecules (CAMs) allowed them to identify specific CAMs, which are required for dendrite bundle organization and may act redundantly. Gain of function (i.e. overexpression) experiments for one of the CAMs, SAX-7, affect the stereotypic pattern observed in the dendrite bundle. Overall, the data is nicely presented and clearly described, although the text could be somewhat tightened up in places. The authors provide a detailed and thorough statistical analysis to describe and quantify the fasciculation of the amphid dendrite bundle, which may be useful in many other instances and for people in other fields.

A concern is, however, that amphid dendrites develop in a rather unusual way, by retrograde extension; i.e. the ciliated endings of the dendrites attach to the tip of the nose and extend through the retrograde migration of the neuronal cell bodies towards their destination near the posterior bulb of the pharynx. This retrograde cell migration could profoundly influence bundle organization. An analysis of cell body position for the cells and genes being studied at different time points may be required to rule out defects resulting of abnormal migration patterns. Conspicuously absent from the analysis is a time course for the *cdh-4* cadherin and an analysis of long isoform *sax-7L* for which specific alleles exist. Thus, it is not clear, which of the observed bundling defects in adults are actually maintenance defects and which are developmental defects. There are also concerns regarding the overexpression experiments of *sax-7*, which was overexpressed in all amphid dendrites. These experiments produced rather subtle defects, which are difficult to interpret. Certainly, something was disturbed, but whether that is sufficient to proof the differential adhesion hypothesis or instructiveness remains less clear. I think an opportunity was missed here. More targeted expression of SAX-7 or CDH-4 in specific cells of the amphid bundle could have allowed more concrete predictions and hypotheses to be tested or, at a minimum provide stronger evidence for an instructive function. In fact, more conclusive data for *sax-7* misexpression in the same amphid neurons but in regard to the effects of *sax-7* on cell body position (and adhesion between cell bodies) was previously described (Sasakura et al., 2005). Finally, the specificity of this SAX-7 misexpression is unclear. Maybe anything expressed (extracellularly or intracellularly) in all amphid dendrites results in subtle defects? Appropriate controls such as for example deletion constructs of SAX-7 or other Ig/FNIII containing proteins would probably have to be tested to make the claim of instructiveness.

Reviewer #2:

In this manuscript, the authors provide some new insights into a largely unexplored area of research, i.e. how do neuronal processes acquire and maintain reproducible positions within a neurite bundle? The authors use *C. elegans*, more specifically the dendrites of amphid sensory neurons, to identify mechanisms that specify dendrite position. First, they developed a workflow to semiautomatically assess the relative position of three dendrites from light-microscopic images, where dendrites were labelled with different fluorescent proteins. They show that in wildtype the dendrite order is conserved but not completely invariant from animal to animal and is largely maintained during larval growth of an individual animal (thus confirming and expanding the results of previous EM studies). Interestingly the order of the dendrites changes from the tip of the nose (the end of the dendrite) to the cell body with a well-defined switch point close to the nose. The authors then show that cilia and amphid sheath cells (somewhat unexpectedly) are not required to establish the order of the dendrites. Finally, the authors screen a list of candidate cell adhesion molecules for defects in dendrite ordering. They found that mutations in the cadherin *cdh-4* randomizes the order of dendrites. Mutations in *ptp-3* and *sax-7* lead to a dendrite order that is distinct from wild type in early larvae and random later, suggesting a role for these adhesion molecules and receptors in establishing the dendrite order.

The manuscript is well written and easy to follow. The experiments largely support the conclusions.

A major claim stated in the Abstract and the Discussion section is: "Our results suggest that differential expression of CAMs allows dendrites to self-organize …" (quoted from the Abstract). However, the authors provide no evidence that the relevant genes (*cdh-4, ptp-3* and *sax-7*) act as adhesion molecules and not as receptors. Furthermore, only one experiment addresses the "differential expression" aspect (ectopic expression of *sax-7*). The results of this experiment, essentially no effect on the order of the dendrites (but a shift in the switch point), seems to support the "glue hypothesis" rather than the differential expression hypothesis. I suggest the authors interpret their results more cautiously.

Figure 2: In the 24h plot (Figure 2B), the order of dendrites close to the nose seems random (light blue colors). At 72h, the order close to the nose is non-random (dark color). This is unexpected, since the dendrites attach to the nose during embryogenesis and stay attached. So, dendrite order at the nose should not change during larval development. The authors should discuss this unexpected observation. One possible explanation is that the distances between dendrites at the nose in young animals are at or below the resolution limit of the microscope used. It would be helpful to have more information about the actual distances between dendrites at various stages and positions. In fact, the authors seem to ignore the resolution limit completely. The processing algorithm will always give a position and distance information. However, this is meaningless information, if the distances are smaller than the resolution limit. In general, the authors should exclude such data from the analysis. I consider this a serious issue potentially compromising some of the data.

Resampling for statistical tests (subsection “Statistical analyses”): The authors used permutation tests to compare populations of wildtype and mutant animals. They present the resulting (501) ranked p-values with a color code (e.g. see Figure 3). It is not clear how the ranking of p-values should be interpreted. Darker colors indicate differences that are non-random, but when does this become meaningful? E.g *mec-8* (Figure 3C) looks more random and less like wt in the middle section of the dendrite (light blue and dark red in the statistical analysis). Is this meaningful? The authors do not explain how they used the ranked p-values to declare whether mutant and wildtype show "significant" differences.

---

## [Author Response]

Reviewer #1:[…]A concern is, however, that amphid dendrites develop in a rather unusual way, by retrograde extension; i.e. the ciliated endings of the dendrites attach to the tip of the nose and extend through the retrograde migration of the neuronal cell bodies towards their destination near the posterior bulb of the pharynx. This retrograde cell migration could profoundly influence bundle organization. An analysis of cell body position for the cells and genes being studied at different time points may be required to rule out defects resulting of abnormal migration patterns.

Altered cell body positioning during retrograde extension would be an interesting mechanism to explain changes in dendrite order, although it would not account for the progressive loss of dendrite order during larval growth. Indeed, *sax-7* has been reported to exhibit mispositioned cell bodies in L1 animals, albeit only with 10% penetrance (Sasakura et al., 2005) that would probably not explain the highly penetrant defects in dendrite order that we observe. Cell body positioning for wild-type, *cdh-4, sax-7*, and *ptp-3* animals has been added to Figure 4—figure supplement 1B and Figure 5—figure supplement 1B and E. Consistent with previous reports, we observed occasional cell body mispositioning in newly hatched *sax-7* animals, however the relative order of the cell bodies remained wild-type (they did not flip positions like the dendrites).

Conspicuously absent from the analysis is a time course for the cdh-4 cadherin and an analysis of long isoform sax-7L for which specific alleles exist. Thus, it is not clear, which of the observed bundling defects in adults are actually maintenance defects and which are developmental defects.

We have added the 24h time point data for *cdh-4* to Figure 4—figure supplement 1C. Dendrites are already randomly ordered at this earlier timepoint. We have also added the *sax-7L*-specific allele, *sax-7(nj53)*, to Figure 5—figure supplement 1G. This allele is somewhat complicated to interpret, as it prevents SAX-7L expression but may also increase SAX-7S expression (Sasakura et al. 2005), but we agree it makes sense to include it. Dendrite order is mostly unaffected.

There are also concerns regarding the overexpression experiments of sax-7, which was overexpressed in all amphid dendrites. These experiments produced rather subtle defects, which are difficult to interpret. Certainly, something was disturbed, but whether that is sufficient to proof the differential adhesion hypothesis or instructiveness remains less clear. I think an opportunity was missed here. More targeted expression of SAX-7 or CDH-4 in specific cells of the amphid bundle could have allowed more concrete predictions and hypotheses to be tested or, at a minimum provide stronger evidence for an instructive function. In fact, more conclusive data for sax-7 misexpression in the same amphid neurons but in regard to the effects of sax-7 on cell body position (and adhesion between cell bodies) was previously described (Sasakura et al., 2005).

We have added additional *sax-7* misexpression experiments, using a promoter strongly expressed in a single neuron (*odr-10*pro, AWA); a promoter expressed broadly in the amphid at low levels (*ptp-3b*pro); as well as the promoter we already included that is expressed broadly in the amphid at high levels (*osm-6*pro). These data are in Figure 6C and suggest a dose-response relationship. However, we appreciate the cautionary note that these experiments do not prove differential adhesion or instructiveness (and do not disprove alternative explanations) and we have revised the text to try to capture this.

Finally, the specificity of this SAX-7 misexpression is unclear. Maybe anything expressed (extracellularly or intracellularly) in all amphid dendrites results in subtle defects? Appropriate controls such as for example deletion constructs of SAX-7 or other Ig/FNIII containing proteins would probably have to be tested to make the claim of instructiveness.

Indeed, we would predict that any construct that modifies cell-surface adhesion could alter dendrite order, including deletion mutants of SAX-7 or other CAMs. We have revised the text to reflect this.

Reviewer #2:[…]A major claim stated in the Abstract and the Discussion section is: "Our results suggest that differential expression of CAMs allows dendrites to self-organize …" (quoted from the Abstract). However, the authors provide no evidence that the relevant genes (cdh-4, ptp-3 and sax-7) act as adhesion molecules and not as receptors. Furthermore, only one experiment addresses the "differential expression" aspect (ectopic expression of sax-7). The results of this experiment, essentially no effect on the order of the dendrites (but a shift in the switch point), seems to support the "glue hypothesis" rather than the differential expression hypothesis. I suggest the authors interpret their results more cautiously.

These points are all well taken. We have revised the Abstract to highlight "combinations of CAMs" (rather than their differential expression) and "altered order" (rather than a new order) and have made similar changes in the Discussion section. We have added additional data to Figure 6 to support the idea that misexpression of *sax-7* produces a subtle but reproducible alteration in dendrite order. We have also reworked the Results section for Figure 6 and parts of the Discussion section to emphasize (i) that our data does not exclude CAM-independent mechanisms that might also contribute to dendrite order (the 'glue' hypothesis) and (ii) that we have not distinguished adhesion vs signaling roles for CAMs.

Figure 2: In the 24h plot (Figure 2B), the order of dendrites close to the nose seems random (light blue colors). At 72h, the order close to the nose is non-random (dark color). This is unexpected, since the dendrites attach to the nose during embryogenesis and stay attached. So, dendrite order at the nose should not change during larval development. The authors should discuss this unexpected observation. One possible explanation is that the distances between dendrites at the nose in young animals are at or below the resolution limit of the microscope used.

This is a good catch; there are several nuggets like this throughout the data that are interesting to speculate about. We hope the online interactive browser will encourage readers to explore them. In this case, the differences are being driven by the 4-6 yellow bins at the nose of the 24h animal. Examining these bundles, we find that the actual pairwise distances are reasonable (~0.4-0.5 µm) so it is unlikely to be a resolution issue, although this was a good guess. We speculate that it may relate to the geometry of the bundle at the nose tip, with younger animals having bundles that are rounder at the nose such that dendrites can engage multiple neighbors at once, and older animals having the bundle squeezed flatter at the nose thus limiting each dendrite to its highest affinity neighbors. But, because it is not a key point and we can only speculate about it, we have chosen not to elaborate on this or other nuggets and to keep our focus on the major conclusions.

It would be helpful to have more information about the actual distances between dendrites at various stages and positions. In fact, the authors seem to ignore the resolution limit completely. The processing algorithm will always give a position and distance information. However, this is meaningless information, if the distances are smaller than the resolution limit. In general, the authors should exclude such data from the analysis. I consider this a serious issue potentially compromising some of the data.

In fact, the resolution limit prevented us from analyzing younger animals (0h, L1 stage) for exactly these reasons. We have added this to Figure 2—figure supplement 1 with a summary of the wild-type average pairwise distances at each developmental stage. Pairwise distances for all genotypes are also now available in the online browser. We had originally considered adding a fourth color (e.g., black) for ambiguous datapoints (using some arbitrary cut-off) but, in practice, sections of the bundle where pairwise distances are essentially equal to each other reveal themselves in the population plots as flickering colors within a column, and they are averaged out in the summary plots and statistics. Indeed, one of the strengths of using our quantitative population-based approach is that it is robust to this kind of noise, and we think its success is demonstrated in the way that the calculated dendrite order is consistent across developmental time points, across a range of genetic perturbations, and agrees well with previous EM analysis.

Resampling for statistical tests (subsection “Statistical analyses”): The authors used permutation tests to compare populations of wildtype and mutant animals. They present the resulting (501) ranked p-values with a color code (e.g. see Figure 3). It is not clear how the ranking of p-values should be interpreted. Darker colors indicate differences that are non-random, but when does this become meaningful? E.g mec-8 (Figure 3C) looks more random and less like wt in the middle section of the dendrite (light blue and dark red in the statistical analysis). Is this meaningful? The authors do not explain how they used the ranked p-values to declare whether mutant and wildtype show "significant" differences.

We have added sentences to the Results section and the Figure 1 legend to clarify that the ranked p-values are equivalent to a corrected p-value (e.g., 25/501 is equivalent to p=0.05). This use of a ranked value statistic is comparable to other commonly used non-parametric ranked tests, for example the Mann-Whitney U-test. There is no cut-off for meaningfulness or significance, but the value of any statistical test is that it allows us to quantitatively compare differences between samples (e.g., to say that the difference between *mec-8* and wild type is less than the difference between *cdh-4* and wild type). We wanted to show the actual values rather than impose an arbitrary cut-off, especially because our data involve differences that vary in their extent along the length of the bundle (for example, compare the red bars in Figure 6D). However, we appreciate that people perceive color shades differently and some readers may find the color bars challenging to interpret; we hope that the online browser will provide an additional resource to explore and compare the p-value rankings directly (e.g., *mec-8* vs wt hovers around p=0.05 for two stretches of about 5 bins each).